



# Wind farm layout optimisation including meandering correction and momentum conserving superposition

Daniel Sukhman[1], Jan Bartl[1], Thomas H. Hansen[1], and Gloria Stenfelt[1]

[1]Western Norway University of Applied Sciences, Inndalsveien 28, 5063 Bergen, Norway

**Correspondence:** Daniel Sukhman (daniel.sukhman@hvl.no)

**Abstract.** This study introduces a combined analytical approach, to simulate the wake flow in large-scale offshore wind farms and forecast their power output. The developed tool, Qwyn, integrates the Ishihara-Qian single-wake model, a momentum-conserving superposition method by Zong & Porté-Agel, and a wake meandering correction by Braunbehrens & Segalini. A validation against both measurement data and Large Eddy Simulation (LES) results at the Horns Rev 1 (HR1) wind farm is performed. This demonstrates the tool's capability in capturing the wind farm flow and power output for different wind directions. Notably, the momentum-conserving superposition method significantly enhances the accuracy of power prediction for narrow directional wind bins, while the wake meandering correction improves precision for wider bins. Subsequently, the validated computational tool is used to optimise the layout of HR1 to enhance its annual power production (AEP). By introducing a convexly shaped layout a projected 0.12% increase in AEP when optimising for a wind rose with 72 wind bins is found. Furthermore, this study shows, that omitting the meandering correction leads to an even more convexly shaped layout without substantial change in AEP improvement.

## 1 Introduction

Wind energy continues to play a pivotal role in global renewable energy strategies, with an ever-increasing focus on optimising wind farm efficiency. The arrangement of turbines within a wind farm significantly impacts its overall performance (Barthelmie et al., 2007). With the rapid growth of offshore wind farms, optimising their layout becomes increasingly complex yet essential for improving the annual energy production (AEP) (Charhouni et al., 2019; Kirchner-Bossi and Porté-Agel, 2018, 2021). Due to the complexity of the case, running such optmisation cases is very time consuming. Analytical modelling approaches present a quick possibility of modelling wake flows within wind farms, which qualifies them for use in optimisation algorithms.

The core wake model used in this work is the Ishihara-Qian model, which considers a Gaussian velocity deficit similarity profile (Ishihara and Qian, 2018). Whilst numerous Gaussian wake models exist (Frandsen et al., 2006; Bastankhah and Porté-Agel, 2014), none of them includes the prediction of rotor-added turbulence. However, this information is crucial for modelling the flow within a wind farm, as the wake recovery is significantly affected by the ambient turbulence levels, which grow within the wind farm (Crespo and Hernández, 1996; van der Laan et al., 2023). This effect can be considered by including a turbulence model such as introduced by Crespo and Hernández (1996). The comparison of modelled wake data to wind tunnel experiments showed that the consideration of rotor-added turbulence intesity increases the modelling accuracy (Polster et al.,





2018). However, the Ishihara-Qian model's ability to predict both velocity deficit and rotor-added turbulence intensity within a single framework presents a significant advantage for practical implementation.

Furthermore, when modeling wake effects on a farm scale, it is essential to consider the overlap of wakes during computation. There are several superposition methods available, which all rely on mathematical intuitive rules (Lissaman, 1979; Katic et al., 1986; Niayifar and Porté-Agel, 2016; Voutsinas et al., 1990). Hereby, the superposition method proposed by Niayifar and Porté-Agel (2016) yields the best results compared to experimental & LES data and is therefore widely used in open-access wind farm modelling tools, such as FLORIS and FOXES (Fleming et al., 2020; Schmidt et al., 2023). This approach will be hereafter referred to as the conservative superposition method. To include a more physical description of a superimposed wake, Zong and Porté-Agel (2020) introduced a new, iterative superposition method derived from momentum conservation. This approach showed lower modelling errors with respect to LES & experimental data and will therefore be used in the present work.

Finally, the traditional wake modelling methodology is extendend by taking first steps towards including the effect of wake meandering. While Medici and Alfredsson (2006, 2008) found evidence for a wake's dynamical meandering motion through large atmospheric eddies, the effect has been long neglected in wind farm modelling. In a further experimental advance, Coudou et al. (2017) showed, that the meandering motion persists within wind farms, influencing their power output. The wake's influence on downstream turbines is known to decrease with more active meandering, which means that modelling approaches neglecting this effect will always tend to overestimate the modelled power losses (Braunbehrens and Segalini, 2019). Since the steady state nature of this studys approach does not allow to resolve time dependent effects, a statistical meandering correction introduced by Braunbehrens and Segalini (2019) is added to the modelling framework.

## 2 Wake modelling & superposition

This section gives insight into the key aspects and interconnection of the modells used. The computational framework computes a plane at hubheight of the modelled wind farm and therefore does not capture the wind's shear profile. All used models support this simplification by neglecting the third dimension as will be shown in the following sections.

### 2.1 Ishihara-Qian wake model

The single wake model used in this work is based on the assumption of a Gaussian-shaped wake, derived from momentum conservation. Since the model also considers rotor-added turbulence intensity, the main equations will be explained in two parts: velocity deficit and rotor-added turbulence intensity. All information given in this section is based on the latest published version of the Ishihara-Qian model (Ishihara and Qian, 2018).

*Velocity deficit*




The basic expression of the model is given by

$$\frac{\Delta u(x,y)}{u_{\text{wind}}} = C(x) \cdot \phi(y), \tag{1}$$

where $x$ and $y$ are the coordinates of a downstream point behind the turbine's hub, $\Delta u$ is the velocity deficit in the wake region, $C$ is the wake-deficit intensity and $\phi$ is the distribution function of the wake's cross-section. Here, $C$ is the streamwise function of the wake shape, which mainly dependends on the downstream distance $x$, while $\phi$ in turn is the spanwise function. Assuming linear expansion of the wake, the wake width $\sigma$ can be expressed with

$$\frac{\sigma}{D} = k\frac{x}{D} + \varepsilon. \tag{2}$$

Here, $D$ is the rotor diameter, $k$ is the wake growth rate and $\varepsilon$ is a linear offset parameter equalling to $\sigma/D$ for $x = 0$. Both values were obtained from fitting LES data resulting in $k$ and $\varepsilon$ being functions of the turbine's thrust coefficient $C_T$ and the ambient turbulence intensity $TI_a$ resulting to

$$k = 0.11 C_T^{1.07} TI_a^{0.20}, \quad \varepsilon = 0.23 C_T^{-0.25} TI_a^{0.17}. \tag{3}$$

Having defined the wake width $\sigma$, the streamwise function $\phi$ follows as

$$\phi = \exp(-\frac{y^2}{2\sigma^2}), \tag{4}$$

with $y$ being the spanwise coordinate. To find an expression for the streamwise function $C$, Ishihara and Qian (2018) make use of the Taylor expansion, since a purely analytical solution showed diverging behaviour in the far wake for some parameter combinations. Therefore, $C$ can be found using two empirical parameters $a$ and $b$ as

$$C(x) = \frac{1}{(a + b \cdot x/D)^2}. \tag{5}$$

The parameters $a$ and $b$ can be written as a function of $k$ and $\varepsilon$ and can in turn be expressed with $C_T$ and $TI_a$ as

$$a = 0.93 C_T^{-0.75} TI_a^{0.17}, \quad b = 0.42 C_T^{0.6} TI_a^{0.2}. \tag{6}$$

Now, Eq. (5) only yields realistic results for the far wake. Thus, the expression needs to be extended by the term $c$, in order to introduce a correction for the near wake with

$$C(x) = \frac{1}{(a + b \cdot x/D + c \cdot (1 + x/D)^{-2})^2}. \tag{7}$$

Again, the expression for $c$ is obtained by data fitting, resulting in

$$c = 0.15 C_T^{-0.25} TI_a^{-0.7}. \tag{8}$$

Inserting Eq. (4) and (7) in Eq. (1) gives the final expression for the normalised velocity deficit

$$\frac{\Delta u}{u_{\text{wind}}} = \frac{1}{(a + b \cdot x/D + c \cdot (1 + x/D)^{-2})^2}$$
$$\cdot \exp(-\frac{y^2}{2\sigma^2}). \tag{9}$$





*Rotor-added turbulence*

The turbulence in the wake of a wind turbine is the combination of the standard deviation of the ambient flow $\sigma_{u,\mathrm{wind}}$ and the rotor-added fluctuation $\sigma_{+\mathrm{rotor}}$, following

$$\sigma_{u,\mathrm{combined}} = \sqrt{\sigma_{u,\mathrm{wind}}^2 + \sigma_{+\mathrm{rotor}}^2}. \tag{10}$$

Note, that $\sigma_{u,i}$ is the standard deviation of the corresponding velocity and should not be confused with the $\sigma$, which is the common variable for the wake width (cf. Eq. (4)). The rotor-added turbulence intensity is therefore defined as

$$TI_{+rotor} = \frac{\sigma_{+\mathrm{rotor}}}{u_{\mathrm{wind}}}, \tag{11}$$

while the total turbulence in the wake region can be obtained from

$$TI_{a,\mathrm{combined}} = \frac{\sigma_{u,\mathrm{combined}}}{u_{\mathrm{wind}}} = \frac{\sqrt{\sigma_{u,\mathrm{wind}}^2 + \sigma_{+\mathrm{rotor}}^2}}{u_{\mathrm{wind}}}. \tag{12}$$

The rotor-added turbulence can be described analogous to the velocity deficit, by a streamwise function $G$, indicating the maximum added turbulence and a spanwise function $\varphi$

$$TI_{+\mathrm{rotor}} = \frac{\sigma_{+\mathrm{rotor}}}{u_{\mathrm{wind}}} = G(x) \cdot \varphi(y). \tag{13}$$

The rotor-added turbulence has its peak at the wake's outer boundaries. LES data showed, that this spanwise distribution can be described with a Gaussian distribution by combining two curves with their initial peaks at the height of the blade tip and later following the outer wake boundary in streamwise distance. The spanwise function $\varphi$ is modelled with

$$\varphi(y) = k_1 \exp\left(-\frac{(y-D/2)^2}{2\sigma^2}\right)$$
$$+ k_2 \exp\left(-\frac{(y+D/2)^2}{2\sigma^2}\right), \tag{14}$$

using the same wake width $\sigma$ as the velocity deficits in Eq. (2). The parameters $k_1$ and $k_2$ are hereby set to 1 and 0 for coordinates outside the region of the blade tips ($y > D/2$), giving the added turbulence a Gaussian shape on the outside of the wake region. For coordinates inside of the blade tips ($y \leq D/2$), the model parameters are set to fit continuity with

$$k_1 = \begin{cases} \cos^2(\pi/2 \cdot (y/D - 0.5)) & y \leq D/2 \\ 1 & y > D/2 \end{cases} \tag{15}$$


$$k_2 = \begin{cases} \cos^2(\pi/2 \cdot (y/D + 0.5)) & y \leq D/2 \\ 0 & y > D/2 \end{cases} \tag{16}$$





Having the spanwise function defined, the streamwise addition $G$ is computed using

$$G(x) = \frac{1}{d + e \cdot x/D}. \tag{17}$$

Similar to $a$ and $b$, the parameters $d$ and $e$ are obtained through data fitting as

$$d = 2.3 C_T^{-1.2}, \quad e = 1.0 T I_a^{0.1}. \tag{18}$$

Here, too, a near-wake correction is added altering Eq. (17) to

$$G(x) = \frac{1}{(d + e \cdot x/D + f \cdot (1 + x/D)^{-2})^2}, \tag{19}$$

with $f$ again being an empirically fitted parameter

$$f = 0.7 C_T^{-3.2} T I_a^{-0.45}. \tag{20}$$

Combining Eq. (13), (14) and (19), gives the final expression for the rotor-added turbulence $TI_{+\text{rotor}}$

$$\begin{aligned}
TI_{+\text{rotor}} = &\frac{1}{(d + e \cdot x/D + f \cdot (1 + x/D)^{-2})^2} \\
&\cdot \left\{ k_1 \exp\left(-\frac{(y - D/2)^2}{2\sigma^2}\right) \right. \\
&\left. + k_2 \exp\left(-\frac{(y + D/2)^2}{2\sigma^2}\right) \right\}.
\end{aligned} \tag{21}$$

## 2.2 Momentum-conserving wake superposition

As mentioned before, the conservative superposition approaches to predict the properties of a mixed wake region, lack a solid theoretical background. Zong and Porté-Agel (2020) argue that none of the available methods is actually conserving momentum. Following this conclusion, they proposed a momentum-conserving superposition method which will be explained in this section. Note that this model only refers to the superposition of velocity deficits and is unsuitable for superimposing rotor-added turbulence intensities. Figure 1 shows a simple example of three aligned turbines to clarify the indication used for

the following equations. The velocity deficit behind each turbine is labelled with $\Delta u_i$, while the velocity itself is described with $u_i$. The inflow velocity of the corresponding $i$-th turbine is described with $u_{0,i}$.

Applying the law of momentum conservation on a turbine's wake flow, the induced thrust force on the wind turbine $F_{T,i}$ can be expressed with

$$F_{T,i} = \rho \int u_i(x,y) \cdot \Delta u_i(x,y) \mathrm{d}y, \tag{22}$$

assuming a two-dimensional plane at hub height with the streamwise coordinate $x$ and spanwise counterpart $y$. Here, $u_i(x,y)$ denotes the velocity in the wake region at a point $p(x,y)$ downstream, and $\Delta u_i(x,y)$ is the corresponding velocity deficit with respect to the turbine's inflow speed $u_{0,i}$ (cf. Fig. 1). Introducing a spatially mean wake convection velocity $u_{c,i}$, the thrust of the turbine can be related linearly to the velocity deficit in the wake

$$F_{T,i} = \rho u_{c,i}(x) \int \Delta u_i(x,y) \mathrm{d}y. \tag{23}$$



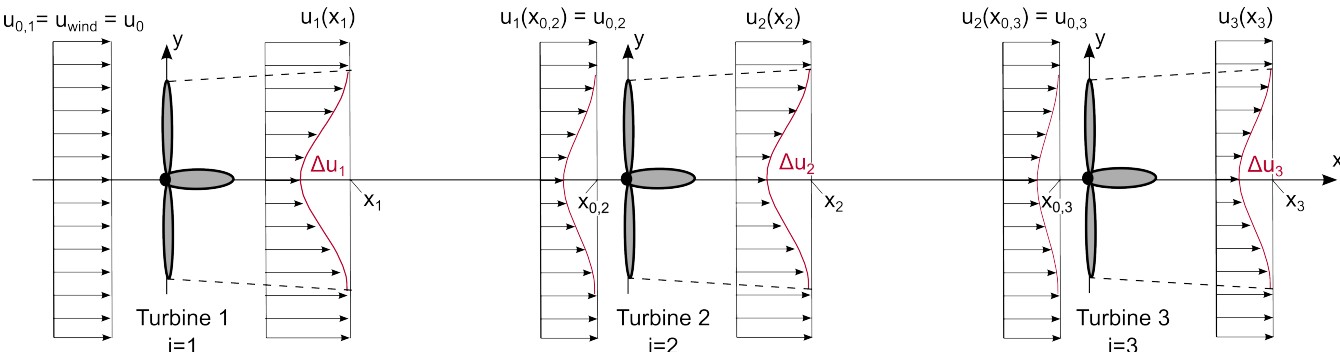

**Figure 1.** Schematic view of velocity profiles and indications for the superposition. Note, that the depicted velocity profiles do not indicate the superimposed result, but rather show the contribution of each turbine displayed.

Substituting Eq. (23) in Eq. (22) yields a mathematical expression for $u_{c,i}$ with

$$u_{c,i}(x) = \frac{\int u_i(x,y) \cdot \Delta u_i(x,y) \mathrm{d}y}{\int \Delta u_i(x,y) \mathrm{d}y}. \tag{24}$$

The convection velocity $u_{c,i}$ is, therefore, a weighted average of the spanwise wake velocity. The explicit expression for $u_{c,i}$, is derived by Zong and Porté-Agel (2020) from the Bastankhah model (Bastankhah and Porté-Agel, 2014) for an $i$-th turbine as

$$\frac{u_{c,i}(x)}{u_{0,i}} = \frac{1}{2} + \frac{1}{2}\sqrt{1 - \frac{C_{T,i}D^2}{8\sigma^2(x)}}, \tag{25}$$

where $u_{0,i}$ is the turbine inflow velocity, $C_{T,i}$ its thrust coefficient, $D$ the turbine diameter and $\sigma(x)$ is the modelled wake width as introduced in Sect. 2.1. Figure 2 shows a schematic view of a modelled wake region with all relevant velocities marked. The

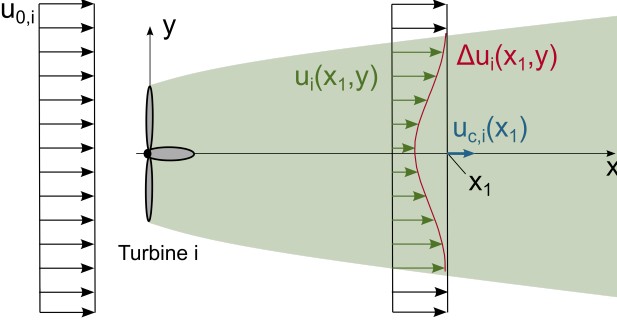

**Figure 2.** Schematic view of an i-th turbine's wake region with inflow velocity $u_{0,i}$, wake velocity $u_i$, velocity deficit $\Delta u_i$ and convection velocity $u_{c,i}$.





introduced procedure can now be applied to a combined wake as it occurs in wind farms, where Fig. 3 gives an overview of all notable velocities. First, applying momentum-conservation to a wind farm gives the sum of all turbine's thrust forces $\sum_i F_{T,i}$ using the total wake velocity $U_{\text{Farm}}$ and deficit $\Delta U_{\text{Farm}}$

$$\sum_i F_{T,i} = \rho \int U_{\text{Farm}}(X,Y) \cdot \Delta U_{\text{Farm}}(X,Y)\mathrm{d}Y. \tag{26}$$

Here, global coordinates $X$ and $Y$ are used and $\Delta U_{\text{Farm}}$ is defined as $U_{\text{Farm}} - u_{\text{wind}}$, which is the difference between the total farm wake velocity and ambient wind speed. When combining Eq. (23) and Eq. (26), it becomes clear that $U_{\text{Farm}}$ has to satisfy

$$\rho \int U_{\text{Farm}}(X,Y) \cdot \Delta U_{\text{Farm}}(X,Y)\mathrm{d}Y = \sum_i F_{T,i}$$

$$= \sum_i \left( \rho u_{c,i}(X) \int \Delta u_i(X,Y)\mathrm{d}Y \right). \tag{27}$$

Analogous to the single wake convection velocity $u_{c,i}$, a global wind farm convection velocity $U_{c,\text{Farm}}$ can be found using the

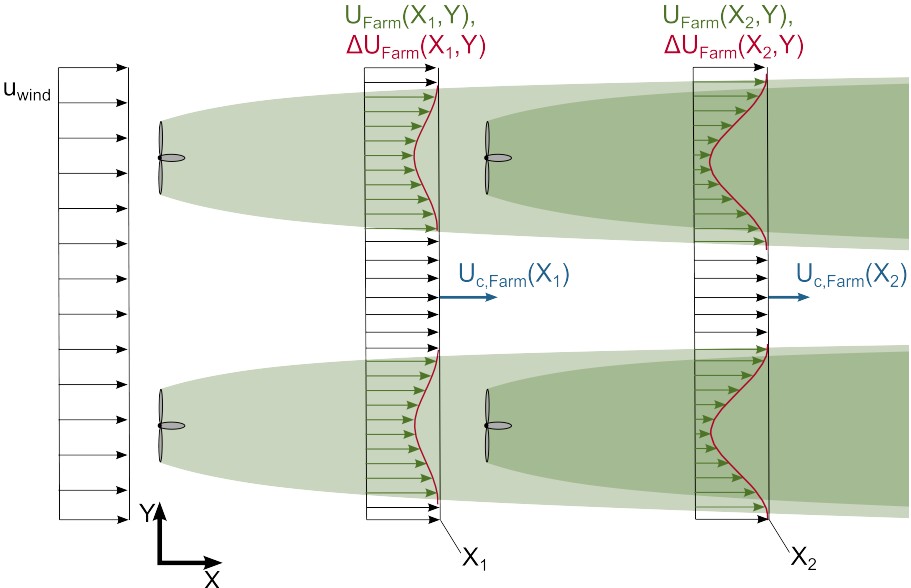

**Figure 3.** Schematic view of a control volume containing a whole wind farm with the ambient velocity $u_{\text{wind}}$, wake velocity $U_{\text{Farm}}$, velocity deficit $\Delta U_{\text{Farm}}$ and convection velocity $U_{c,\text{Farm}}$.

global values of wake velocity and deficit with

$$U_{c,\,\text{Farm}}(X) = \frac{\int U_{\text{Farm}}(X,Y) \cdot \Delta U_{\text{Farm}}(X,Y)\mathrm{d}Y}{\int \Delta U_{\text{Farm}}(X,Y)\mathrm{d}Y}. \tag{28}$$





This expression can be rewritten, discretising it to match the explicit, analytical nature of the used modelling approach as

$$U_{c,\,\mathrm{Farm}}(X_j) = \frac{\sum_Y U_{\mathrm{Farm}}(X_j, Y) \cdot \Delta U_{\mathrm{Farm}}(X_j, Y)}{\sum_j \Delta U_{\mathrm{Farm}}(X_j, Y)}, \qquad (29)$$

with index $j$ indicating each considered global streamwise position $X_j$ as displayed in Fig. 3. In other words, a global convection velocity $U_{c,\,\mathrm{Farm}}(X_j)$ is obtained by computing the weighted average of the spanwise farm wake velocities. Finally, the conservation of momentum during the superposition of wakes is considered by computing $\Delta U_{\mathrm{Farm}}$ with

$$\Delta U_{\mathrm{Farm}}(X, Y) = \sum_i \frac{u_{c,i}(X)}{U_{c,\,\mathrm{Farm}}(X)} \Delta u_i(X, Y). \qquad (30)$$

This indicates that the combined velocity deficit of several overlapping wakes at a global point $p_{\mathrm{global}} = (X, Y)$ can be computed as a weighted sum of the wake deficits induced by each turbine at this very point. The weighting factor here is the ratio of the individual convection velocity $u_{c,i}$ and the farm's global convection velocity $U_{c,\,\mathrm{Farm}}$. Since $U_{c,\,\mathrm{Farm}}$ is computed using values for the combined wake velocity and deficit, it has to be computed iteratively. For the first iteration, an initial value of $U_{c,\,\mathrm{Farm}}^{\mathrm{initial}}$ is approximated by evaluating the highest available individual convection velocity at each given point as

$$U_{c,\,\mathrm{Farm}}^{\mathrm{initial}}(X_j) = \max(u_{c,i}(X_j)). \qquad (31)$$

For all further iterations Eq. (29) is used to compute $U_{c,\,\mathrm{Farm}}$.

## 2.3 Meandering correction

Wake meandering has been known for almost as long as efforts for turbine wake characterisation have been made (Baker and Walker, 1984). Yet, the lack of data quantifying the meandering motion has made it difficult to incorporate it into wake modelling and power prediction. Larsen et al. (2008) proposed a first approach for a numerical dynamic wake meandering model for CFD applications. To include this effect in analytical modelling Braunbehrens and Segalini (2019) introduced an approach to express the meandering motion of the wake as a time-averaged, explicit function using statistical meandering data from numerically solving linearised Navier-Stokes equations. This section gives an overview of the model presented by Braunbehrens and Segalini (2019) and connects it to the Ishihara-Qian wake model presented in Sect. 2.1.

The main assumptions for the statistical wake meandering model are, firstly, that the wake is a passive tracer, not influencing the ambient flow, and secondly, that the wake maintains its basic shape while moving in the lateral and vertical directions (Braunbehrens and Segalini, 2019). Following these thoughts, a time-averaged, meandering velocity deficit profile $\Delta u_{i,m}$ can be drawn as shown in Fig. 4. As can be seen, the lateral movement of the wake causes the effective wake width to be wider than a stable wake's width. At the same time, the intensity of the velocity deficit decreases. This also applies to the height profile of the wake, since the meandering motion occurs in both, horizontal and vertical directions.

Now, following this introduction and keeping the equation for the wake velocity deficit in mind (cf. Eq. (9)), two alterations are to be made to the initial modelling approach. First, the streamwise function $C$ as presented in Eq. (5) has to be reduced, while the width of the spanwise function $\phi$ (cf. Eq. (4)) is to be increased. For this reason, Braunbehrens and Segalini (2019)





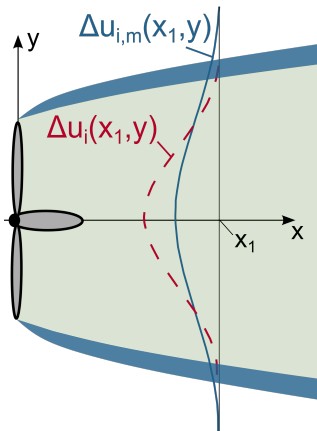

**Figure 4.** Qualitative velocity deficit profile of a turbine wake with $\Delta u_{i,m}$ (solid line) and without time averaged wake meandering $\Delta u_i$ (dashed line), inspired by de Vaal and Muskulus (2021). The dark area indicates the added wake width in comparison to the light-coloured area of a non meandering wake.

introduce the statistic parameter $\sigma_{m,i}$ describing the standard deviation of the wake centre[1], which can be computed for the $i$-th turbine with

$$
\sigma_{m,i}(x)^2 = 2\psi\Lambda^2 \left[ \frac{x}{u_{c,i}(x)\Lambda} + \exp\left( \frac{-x}{u_{c,i}(x)\Lambda} \right) - 1 \right].
$$
(32)

Here, $\psi$ is the fluctuation intensity of the wake, $\Lambda$ is the integral length scale of the meandering inducing turbulent eddies, $x$ is the streamwise distance, while $u_{c,i}$ is the already known convection velocity introduced in Sect. 2.2 (cf. Eq. (25)). The fluctuation intensity $\psi$ depends on the ambient conditions and can either be computed using the friction velocity $u_*$ (Braunbehrens and Segalini, 2019) with

$$
\psi = 2.5\,u_*,
$$
(33)

or using the ambient conditions of the free flow (de Vaal and Muskulus, 2021) as

$$
\psi = 0.7\,TI_a \cdot u_{\text{wind}}.
$$
(34)

Both approaches initially yield very similar results. However, while Eq. (33) is purely derived from surface roughness, which is not constant in offshore conditions (Edson et al., 2013), the second approach is using the ambient wind speed and turbulence intensity. When applied to the $i$-th turbine within a wind farm, Eq. (34) can be rewritten as

$$
\psi_i = 0.7\,TI_{0,i} \cdot u_{0,i}.
$$
(35)

---

[1]Not to be confused with the wake width $\sigma$ or the standart deviation of wind velocity $\sigma_{u,i}$ introduced in Sect. 2.1.



By using the real inflow conditions $u_{0,i}$ and $TI_{0,i}$, $\psi_i$ accounts for the changing inflow conditions within the wind farm.
Therefore, in this work, $\psi$ is computed as shown in Eq. (35). Now, Braunbehrens and Segalini (2019) provide an assumption
for the integral length scale. They suggest $\Lambda$ to be chosen with respect to the height $z$ above the ground as

$$\Lambda = \frac{\kappa z}{\psi}, \tag{36}$$

where $\kappa \simeq 0.4$ is the Karman constant. This approach is motivated by the expectation, that the meandering causing eddies
increase in size with height. They further suggest to use the hub height of the considered turbine to be $z \simeq z_{\text{hub}}$. Despite this,
the authors of the model firmly point out that investigating $\Lambda$ and its connection to atmospheric conditions by relating it to
LES or weather mast data is crucial when using the presented approach. However, the meandering model is based on data of
the wind farm Horns Rev 1, which is also the focus of this work's study (cf. Sect. 4). As a result, $\Lambda = 0.8$ was chosen for this
work, which reflects the order of magnitude suggested by the model's authors, as well as by LES simulations results (Keck
et al., 2014).

After discussing the central parameter $\sigma_m$ (cf. Eq. (32)), the wake meandering model must be linked to the single wake
model. Directly applying Braunbehrens & Segalinis proposal to the basic equation Eq. (9) of the Ishihara-Qian model yields
the corrected version for the speed deficit $\Delta u_m$

$$\frac{\Delta u_m}{u_{\text{wind}}} = C \left[ 1 + \left( \frac{\sigma_m}{\sigma} \right)^2 \right]^{-1/2} \cdot \exp \left( -\frac{y^2}{2(\sigma^2 + \sigma_m^2)} \right). \tag{37}$$

The authors of the meandering model do not propose how the profile of rotor-added turbulence is to be corrected respectively.
However, as the added turbulence is assumed to be strongly linked to the shape of the wake's velocity profile (cf. Sect. 2.1), the
meandering-corrected expression for the rotor-added turbulence $TI^m_{+\text{rotor}}$ is assumed to be analogous to Eq. (37) and results
from incorporating $\sigma_m$ into Eq. (21) yielding

$$\begin{aligned}
TI^m_{+\text{rotor}} = &G \left[ 1 + \left( \frac{\sigma_m}{\sigma} \right)^2 \right]^{-1/2} \\
&\cdot \left\{ k_1 \exp \left( -\frac{(y - D/2)^2}{2(\sigma^2 + \sigma_m^2)} \right) \right. \\
&\left. + k_2 \exp \left( -\frac{(y + D/2)^2}{2(\sigma^2 + \sigma_m^2)} \right) \right\}.
\end{aligned} \tag{38}$$

## 3  Computation methodology

The models described in Sect. 2 are joined together as Qwyn (quick wake analytical modelling), a computational framework
for wind farm power prediction. This version of the code is written in MATLAB (R2022b) and contains a wind farm model
computing the mean and turbulent flow field in a two-dimensional plane at hub height. Figure 5 shows the main in- and
output parameters of Qwyn. On farm level, the number and position of the turbines is passed on to the code, while data of
the considered turbine model consists of rotor diameter, power- and thrust coefficient. Finally, information on the ambient
flow contains the ambient turbulence intensity, wind speed and wind direction. Qwyn computes the mixed wake region and





estimates the turbine's inflow conditions iteratively. The output consists of an estimate of each turbine's inflow wind speed, turbulence intensity, their power output, as well as the combined power production of the farm for the modelled steady case.

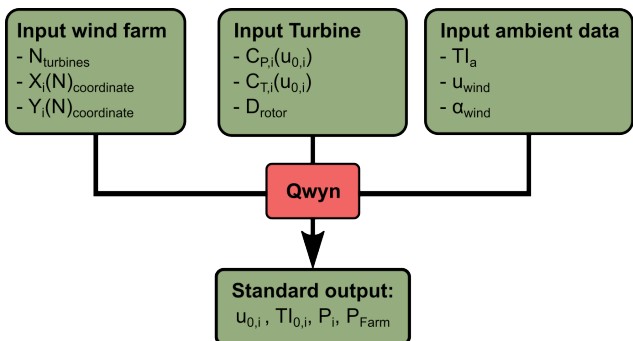

**Figure 5.** In- and output parameters of Qwyn.

To ensure numerical grid independence, a sensitivity study of three main parameters has been conducted. The inflow conditions of each turbine are obtained by averaging $N_{Rotor} = 100$ evenly spaced points in each rotor plane. Further, the computation of a wind farms streamwise convection speed (cf. Eq. (29)) has to be discreticised in lateral direction to the freestream. Here, a number of $N_{U_c} = 200$ points has been found to yield accurate results while being considerate of the computation time. Finally, as the computation is iterative, a termination criterion is defined through

$$\zeta = max(|u_{0,i\,\mathrm{new}} - u_{0,i\,\mathrm{old}}|). \tag{39}$$

Here $\zeta$ is the change in estimated inflow velocity of each turbine from one iteration to another. The value $\zeta = 10^{-3}$ is set with respect to the models accuracy. The relative accumulated numerical error found with respect to the converged quantity is $< 0.053\%$ and therefore negligible.

The computation method presented so far predicts a wind farms power output for a fixed combination of ambient wind speed, turbulence intensity and wind direction. To predict a farm's AEP, all combinations of the mentioned parameters need to be computed and weighted according to ambient data provided by the corresponding wind rose. The tool's input is therefore extended by wind rose data, which consists of $j$ combinations of wind speed and direction, complemented by the according frequencies $f_j$. Each of these combinations needs to be evaluated separately by Qwyn. The according computation scheme is shown in Fig. 6. It can be seen, that the output changes to a series of predicted power for both, the individual turbines $P_{i,j}$, as well as the farms combined power generation $P_{\mathrm{Farm},j}$ for all combinations $j$ of wind rose data. These are then weighted by the according frequency and multiplied with the annual production time $t_{\mathrm{Prod}}$ to obtain the turbines and farms weighted energy production, $EP_{i,j}$ and $EP_{\mathrm{Farm},j}$. Summing up the weighted energy contributions finally yields each turbine's $AEP_i$ and cummulated farm's $AEP_{Farm}$.



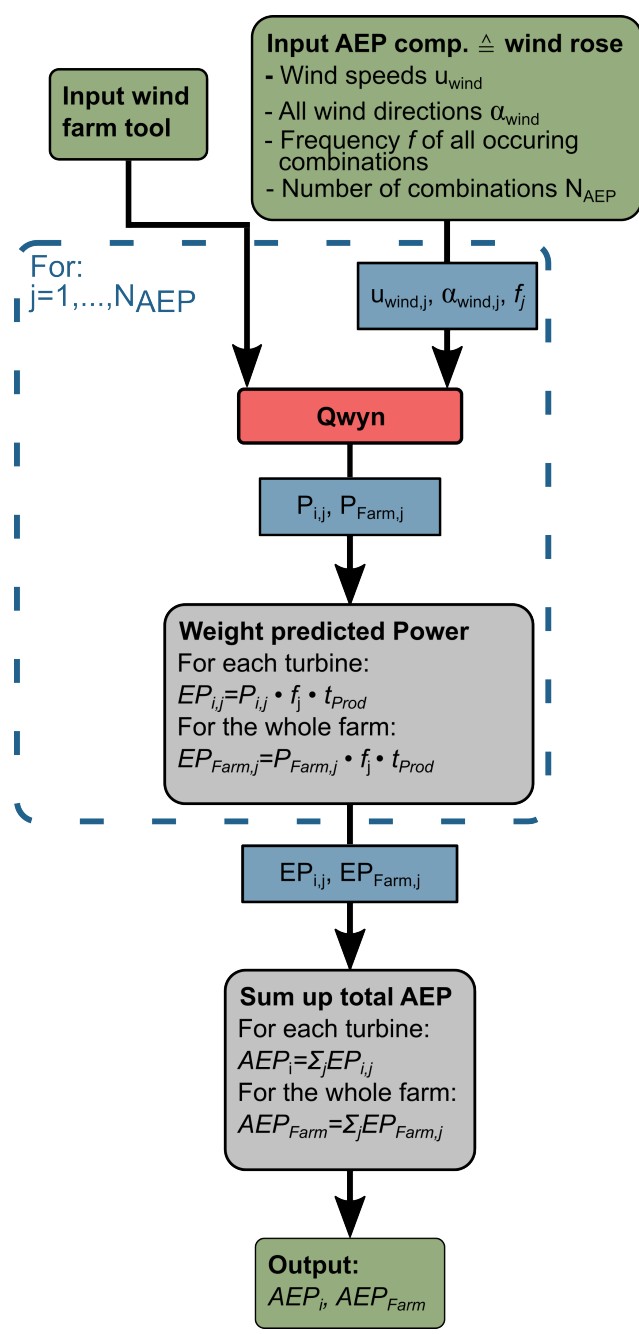

**Figure 6.** Schematic view of the computational flow for a wind farm's AEP prediction.





## 4 Study case

To investigate the validity of the presented approach, a study is performed comparing the modelled results to experimental data, numerical results and other analytical modelling approaches. For this purpose the wind farm Horns Rev 1 (HR1), located in the North Sea in front of the Danish western shore, is considered. It consists of 80 turbines of the type Vestas V80-2MW, whose corresponding power and thrust curve is given in Fig. 7. The farms layout can be seen in Fig. 8. The depicted wind directions

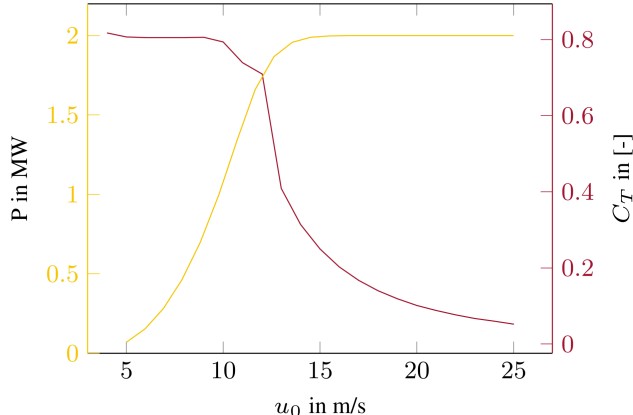

**Figure 7.** Power curve and thrust coefficient over inflow wind speed for the turbine Vestas V80-2MW adapted from Qian and Ishihara (2020).

are worst case operating scenarios, in which most turbines are fully impinged by upstream turbine wake regions, leading to
high wake overlap and power losses. The turbines are placed with a streamwise and spanwise spacing of $S_{x,0} = S_{y,0} = 7\mathrm{D}$ when viewed from the first predominant wind direction $\alpha_0 = 270°$. For $\alpha_1 = 222°$ and $\alpha_2 = 312°$, the corresponding streamwise spacings are $S_{x,1} = 9.4\mathrm{D}$ and $S_{x,2} = 10.4\mathrm{D}$, respectively (Barthelmie et al., 2009). The primary key data of the presented wind farm are summarised in Table 1. The measurement data for this validation case is taken from Barthelmie et al.

**Table 1.** Key data of Horns Rev 1 (Vattenfall, 2023).

| Number of turbines | $N$ | 80 |
|---|---|---|
| Turbine model | - | Vestas V80-2MW |
| Rated power | $P_{\mathrm{rated}}$ | 160 MW |
| Rated wind speed | $u_{\mathrm{rated}}$ | 10 m/s |
| Main wind directions | $\alpha_{0-2}$ | $270°, 222°, 312°$ |
| Corr. streamwise spacing | $S_{x,i}$ | 7D, 9.4D, 10.4D |

(2007, 2009, 2010), who evaluated various data sets for the wind farm's power production at different wind directions. The
data is mostly based on a wind speed of $u_{\mathrm{wind}} = 8\,\mathrm{m/s}$ at an ambient turbulence intensity of $TI_a = 8\%$ with an uncertainty

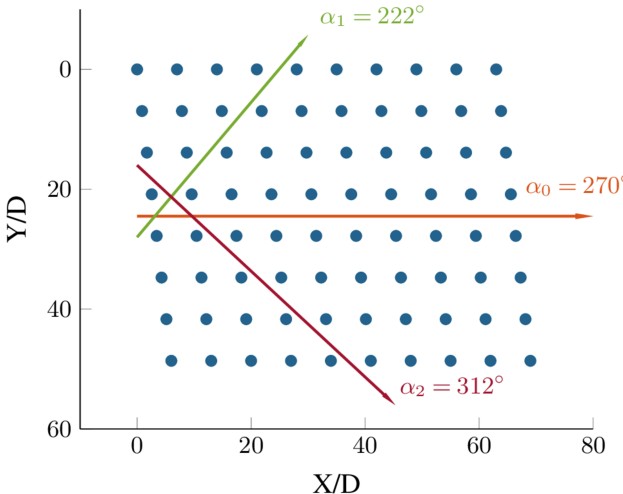

**Figure 8.** Layout of the wind farm Horns Rev 1 with three worst case operation wind directions, adapted from Qian and Ishihara (2020).

of $\pm 0.5$ m/s and $\pm 2\%$, respectively. The wind direction varies in the whole interval from $0°$ to $360°$, as the data sets were obtained over several months. Therefore, the measured power production results from averaging data within certain directional sectors, the so-called wind bins. The following validation study focuses on two bin sizes, $\alpha = \pm 1°$ and $\alpha = \pm 5°$. A summary of the measurement data and the known uncertainties are given in Table 2.

The validation is also complemented by LES results published by Wu and Porté-Agel (2015). Their simulation study, too, focuses on the power production of HR1 at the main wind directions $\alpha_{0-2}$, analogous to the previously mentioned measurement campaign. However, only the bin size of $\alpha \pm 1°$ was considered. All results were obtained by assuming a wind speed of $u_{\mathrm{wind}} = 8$ m/s and an ambient turbulence level of $TI_a = 7.7\%$ for all wind directions. The choice of ambient turbulence corresponds to measurement results presented by Hansen et al. (2012), who showed that the mean turbulence level at HR1 is
nearly the same for all significant wind speeds of $u_{\mathrm{wind}} = 6 - 12$ m/s.

Finally, the analytical results of a conservative approach by Sukhman et al. (2023) are used for direct comparison. These results were likewise obtained using the Ishihara-Qian wake model. However, wake mixing was predicted conservatively, and no additional models (such as wake meandering correction, nor momentum conservation) were considered.

Analogous to the LES and analytical reference data, all results presented in the validation study are computed assuming
$u_{\mathrm{wind}} = 8$ m/s and $TI_a = 7.7\%$, focussing on neutral atmospheric conditions, since this is the predominant state at HR 1 (Barthelmie et al., 2007). A summary of the essential data used for the validation is given in Table 2.

## 5    Validation results

To perform a comprehensive comparison, three versions of Qwyn's output are evaluated to understand the contribution of each implemented model addition. All versions use the Ishihara-Qian single-wake model and vary the following:





**Table 2.** Data and uncertainties of the measurements by Barthelmie et al. (2007, 2009) and corresponding parameters used for this validation case.

| Measurement data | | |
| --- | --- | --- |
| Wind speed | $u_{\mathrm{wind}}$ | $8 \pm 0.5\,\mathrm{m/s}$ |
| Turbulence intensity | $TI_a$ | $8 \pm 2\,\%$ |
| Wind directions | $\alpha_i$ | $222°,\ 270°,\ 312°$ |
| Atmospheric stability | - | neutral |
| Data uncertainty | $\Delta P_i/P_1$ | $\pm 0.1$ |
| **Modelling & simulation data** | | |
| Wind speed | $u_{\mathrm{wind}}$ | $8\,\mathrm{m/s}$ |
| Turbulence intensity | $TI_a$ | $7.7\,\%$ |
| Wind directions | $\alpha_i$ | $222°, 270°, 312°$ |
| Wind bins evaluated | $\Delta\alpha$ | $\pm 1°, \pm 5°$ |
| Atmospheric stability | - | neutral |

1. using **(a) momentum conserving superposition** without **(b) wake meandering correction**,

2. conservative superposition, including **(b) wake meandering correction**,

3. combining both **(a) momentum conserving superposition** and **(b) wake meandering correction**.

The prediction is first performed in 1-degree steps. Afterwards, the obtained results are averaged to account for different sized of wind direction bins.

## 5.1   Power prediction

The results for the inflow case $\alpha_{\mathrm{wind}} = 270° \pm 1°$ can be seen in Fig. 9a. The graphic shows one turbine row's measured, simulated and modelled power output. The conservative approach and the momentum-conserving superposition under-predict the measured and LES results. This aligns with the tendency of analytical models to underestimate a farm's power as statet by Qian and Ishihara (2020). The momentum-conserving superposition seems to show better agreement when compared to the
conservative approach. It follows the trend given by the LES data, resulting in a positive power recovery between the second and third turbine. This suggests that the physical background of the momentum-conserving approach outperforms the conservative approach. On the other hand, the wake-meandering correction leads to over-prediction of the power yield. Qwyn shows an even higher disagreement when combining wake meandering and the new superposition approach. The trend of predicting a higher power output with meandering correction than with the previous models is understandable as adding wake meandering





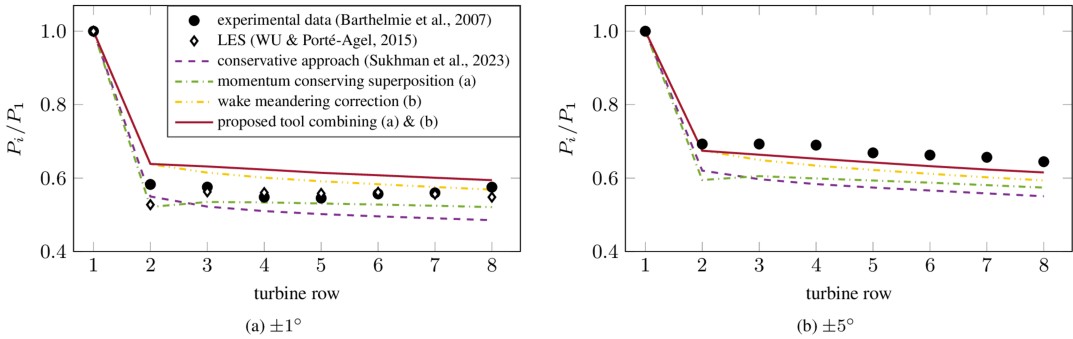

**Figure 9.** Modelled power for one row of HR1 evaluated with (a) $\pm 1°$ and (b) $\pm 5°$ bins. Inflow case: $\alpha_{\mathrm{wind}} = 270°$, $S_x = 7\,\mathrm{D}$.

correction results in a less intense wake and thus lowering the velocity deficits within the farm. Analysing the inflow velocities $u_{0,i}$ of the individual turbines confirm these trends, which is expected since $u_{0,i}$ is the only modelled parameter influencing the power prediction.

The wind direction $\alpha_0 = 270°$ considered so far, is a worst-case scenario for the presented wind farm, as all turbine wakes align within a row of turbines. When considering a wider directional bin of $\pm 5°$, the power prediction will naturally be higher than the values for a narrow bin of $\pm 1°$, as more favourable wind directions are included in the power computation. The wake-meandering model is empirically fitted to a measurement data set based on a wind bin of $\pm 5°$. Therefore, the validation is extended by this wind sector. Figure 9b shows the modelled and measured power prediction for the inflow case $\alpha_{\mathrm{wind}} = 270° \pm 5°$. For this case, however, no LES reference data are available. It is evident that all models under-predict the measured values when modelling a larger bin width. While all results follow the general trend of the validation data, the meandering corrected values show the best performance. This suggests that the combined approach is better at capturing wider bins, while a model neglecting meandering correction leads to better agreement when focusing on narrow bins.

Having understood Qwyn's performance for one wind direction, two more are examined with $\alpha_1 = 222°$, corresponding to $S_x = 9.4\,\mathrm{D}$ and $\alpha_2 = 312°$, corresponding to $S_x = 10.4\,\mathrm{D}$. Both cases are shown in Fig. 10, for $\pm 1°$ on the left-hand side and $\pm 5°$ on the right-hand side. Again, the models considering wake meandering show a significantly better agreement for the wider bin, with a root-square mean error (RSME) of down to $0.60\,\%$. The conservative approach and the momentum-conserving superposition show comparably good agreement for narrow wind bins, with an RSME between $2.52\,\% - 3.71\,\%$. While the model's described trends are reasonable, note that the measurement results have a significant error margin of $\Delta P_i / P_1 = \pm 0.1$ (cf. Table 2).

### 5.2 Sensitivity to wind direction

Since the modelling approach aims to predict a wind farm's AEP, the validation is extended to investigate Qwyn's sensitivity to a wide interval of wind directions. Contrary to the results presented so far, no measurement data are available for comparison. However, Porté-Agel et al. (2013) investigated the performance of HR1 in the directional interval of $\alpha_i = [173°, 354°]$ using





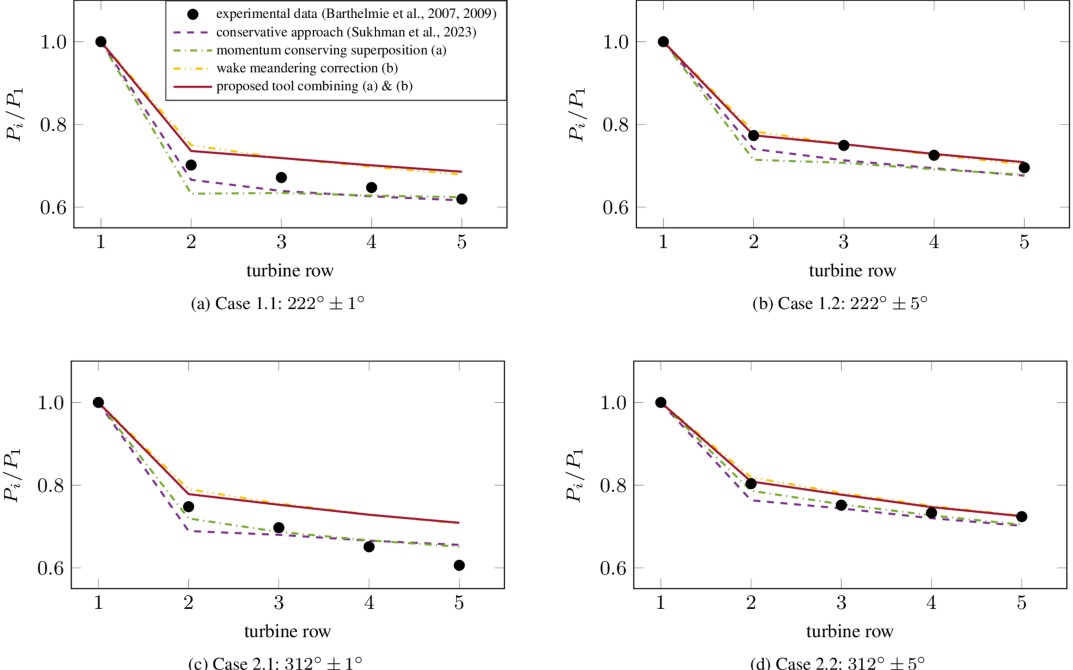

**Figure 10.** Measured and modelled power loss for the wind directions $\alpha_1 = 222°$, $S_x = 9.4\,\mathrm{D}$ (a, b) and $\alpha_2 = 312°$, $S_x = 10.4\,\mathrm{D}$ (c, d). The left-hand side graphs consider a direction bin of $\pm 1°$, while the right-hand side shows a bin of $\pm 5°$.

LES simulations. Additionally, Niayifar and Porté-Agel (2016) performed a comparative study using analytical modelling by implementing the Bastankhah single wake model (Bastankhah and Porté-Agel, 2014) and conservative superposition. Unlike the previous validation, no averaging within wind direction bins is performed in this last study. The reason for that is that isolating the results for each wind direction gives a better understanding of Qwyn's sensitivity. This study's key data are summed up in Table 3.

**Table 3.** Key data of the wind direction sensitivity study.

| Wind speed | $u_{\mathrm{wind}}$ | $8\,\mathrm{m/s}$ |
|---|---|---|
| Turbulence intensity | $TI_a$ | $7.7\,\%$ |
| Wind directions | $\alpha_i$ | $[173°, 354°]$ |
| Atmospheric stability | - | neutral |

Figure 11 shows the power output of the whole farm $P_{\mathrm{Farm}}$ over different inflow angles, normalized by the power output of 80 corresponding turbines operating in the freestream. It can be seen that all displayed analytical models capture the general trends proposed by the LES. However, the models neglecting the wake-meandering correction show better agreement at





inflow directions with high wake interactions. This is reasonable, as the wake meandering correction performs worse when no averaging of wind bins is done during modelling.

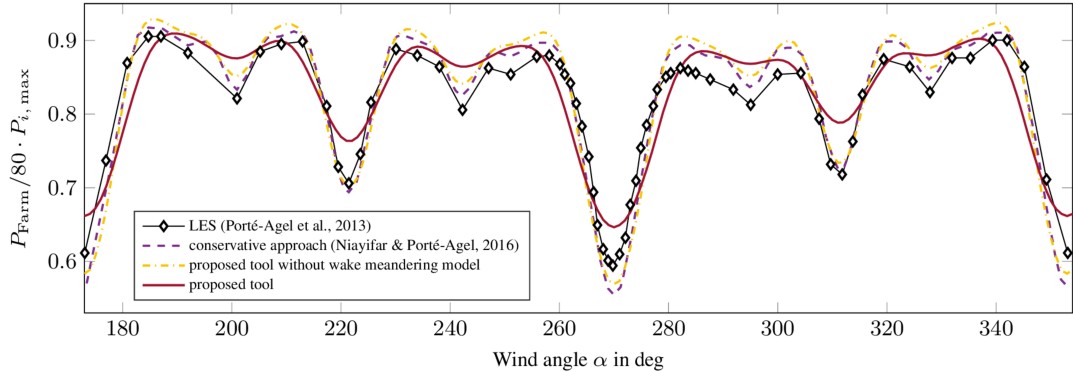

**Figure 11.** Normalised power output of HR1 for a wide range of inflow angles.

All analytical results show some disagreement with the LES data outside of the worst-case scenarios. Two preliminary assumptions can be made to explain this effect. When a wind turbine rotates, its wake is obeying momentum conservation by rotating in the opposite direction. This rotation is amplified, when two wakes are fully overlapping, which is captured by all presented analytical approaches. However, when partially overlapping, the rotational directions of the wakes become opposite, causing the combined turbulence intensities to weaken at the corresponding mixing area (Qian and Ishihara, 2020). None of the analytical approaches used in this study accounts for said effect. This arguably drives the deviations between LES data and analytical results for wind directions where partial wake overlap is dominant. This could be investigated by extending Qwyn to account for partial wake mixing similarly to the approach by Qian and Ishihara (2020).

Secondly, the question may arise if analytical modelling becomes insensitive towards large turbine spacings. As we know, the streamwise distance $S_x$ of the turbines changes with the inflow direction. The smallest distance of $S_{x,0} = 7\mathrm{D}$ corresponds to a inflow direction of $\alpha_0 = 270°$, while the highest value of $S_{x,4} = 28.9\mathrm{D}$ occurs at $\alpha_4 = 284°$. On a first look Fig. 11 can give the impression, that deviation between LES and analytical results is higher for wind directions corresponding to larger turbine spacings. This potential effect has been studied through comparing the tool's modelling results at different turbine spacings to LES data by Porté-Agel et al. (2013). This investigation, however, could not confirm a correlation between modelling accuracy and turbine spacing, which excludes it as a systematic source of error.

### 5.3 Conclusions from validation

The validation study presented confirms the capability of Qwyn to capture the main trends of power yield of the wind farm HR1. Neglecting the introduced meandering correction leads to a reasonable agreement when considering narrow wind bins, yielding a maximum RSME of $3.63\,\%$ with respect to the measurement data. However, when evaluating wider bins adding the meandering correction improves the RSME up to $0.62\,\%$. The proposed tool is sensitive to a wide range of wind directions





($\alpha_i = 173 - 354°$), capturing all main trends as suggested by the reference LES data. Even though the RSME to the validation data is not negligibly small, the tool is well suited for a wind farm's power prediction. One reason for the deviation

is the uncertainty of the validation data used for this study. Firstly, the measurement data have a significant error margin of $\Delta P_i/P_1 = \pm 0.1$. Secondly, several simplifications and follow-up additions have been made to the used reference LES data over time. While some results were sharpened, these changes were not applied to all LES datasets used for the validation. Therefore, the study was performed with respect to the partially outdated, original LES results of Porté-Agel et al. (2013) and Wu and Porté-Agel (2015). Even though an exact quantification of the absolute error is difficult, all shown results picture sim-

ilar trends and predict the same order of magnitude, which supports the informative value of both this work and the reference data.

In addition to the presented study, some further influencing factors still need to be investigated in future. The validation case considers only the wind farm HR1. A comprehensive investigation of Qwyn's sensitivity to different layouts and turbine models should be conducted to confirm the tool's applicability to other farm designs. Further, the validation case assumed

neutral atmospheric stability, a wind speed of $u_{\mathrm{wind}} = 8\,\mathrm{m/s}$ and a turbulence intensity level of $TI_a = 7.7\,\%$. Even though this corresponds well to the average operation condition of HR1, modelling other wind farm locations will require a wider scope of these parameters. Nevertheless, this study shows, that the tool is applicable and valid for HR1, outperforming conservative modelling approaches.

## 6 Layout Optimisation

With the rising availability of computationally inexpensive wind farm modelling approaches, research started to focus on investigating optimal farm layout designs. By considering only one wind direction, Charhouni et al. (2019) showed that the power yield of HR1 could be increased by up to $\Delta P_{\mathrm{Farm}} = 57.89\,\%$. This order of magnitude was confirmed by Sukhman et al. (2023), suggesting a power gain of $\Delta P_{\mathrm{Farm}} = 57.17\,\%$. Both studies showed the potential of a layout optimised towards wake effects. However, the actual operating conditions of a wind farm consist of various wind speeds and directions. Considering

a full wind rose significantly increases the degrees of freedom of this optimisation problem, making it difficult to find an optimum. Kirchner-Bossi and Porté-Agel (2018) took this fact into account and proposed a solution considering 72 different wind directions, which led to an AEP gain of $\Delta AEP_{\mathrm{Farm}} = 0.24\,\%$. This significantly lower power gain is more reasonable when accounting for the complexity of the given optimisation case. All mentioned references modelled the wind farm using different single-wake models and combined them with conservative superposition. This work's layout optimisation focuses on

finding a solution using Qwyn. The following sections first give an overview of the optimisation's computational structure and the reference case used for this study. Then, two optimised layout proposals are compared.

### 6.1 Optimisation set-up

For this study, Qwyn is embedded into an optimisation structure, as shown in Fig. 12, where the gradient based optimisation function *fmincon* (Mathworks, 2023) is used to find the optimal AEP by varying the farm's layout. The input is extended




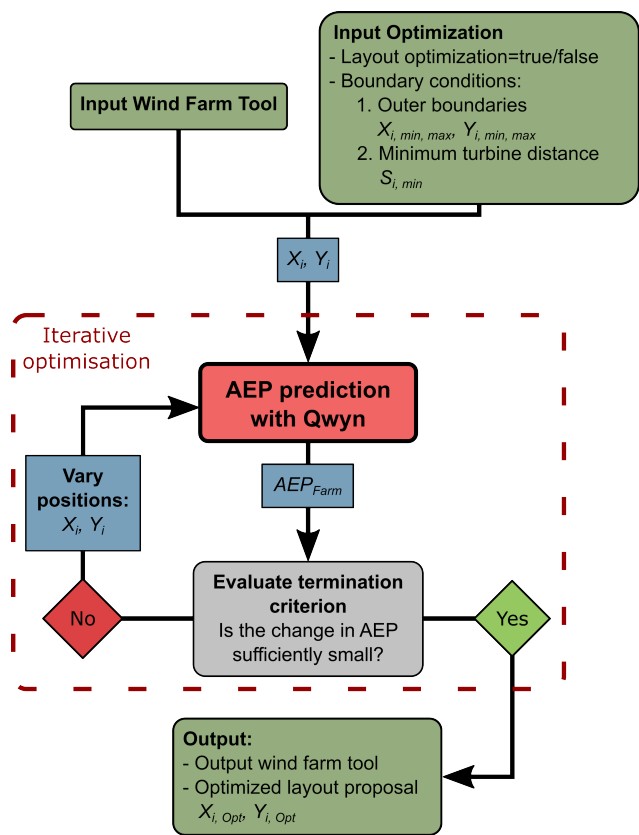

**Figure 12.** Schematic view of the layout optimisation's computation flow.

by boundary conditions, restraining the optimisation problem to a realistic scope. First, the outer boundaries of the modelled domain have to be chosen by specifying a value for $X_{\min,\max}, Y_{\min,\max}$, limiting the farm's outer shape. Next, the minimal turbine distance $S_{\min}$ has to be defined. This value is the radius of a safety space around every single turbine as visualised in Fig. 13. The placement of one turbine inside another turbine's safety space needs to be excluded by the optimiser. If this rule is violated, a turbine could operate in another ones near wake region. This would lead to substantial structural loads, as the near wake is characterised by strong velocity gradients and turbulent structures. The turbine's positions, expressed as global coordinates $X_i$ and $Y_i$, are now used as the input variables for the optimisation problem. A specified initial layout is used as the starting point for the optimisation problem. While trying to find the global optimum, this starting point has to be varied, running repeated optimisations. Here, a handful of random starting points are chosen to investigate the validity of the optimum found. The optimisation function then uses the *interior-point* algorithm, to optimise the target value $AEP_{\mathrm{Farm}}$ by varying the wind farm layout. Once $AEP_{\mathrm{Farm}}$ reaches an optimum, the corresponding layout $X_{iOpt}, Y_{iOpt}$ is given as an additional output of the tool.





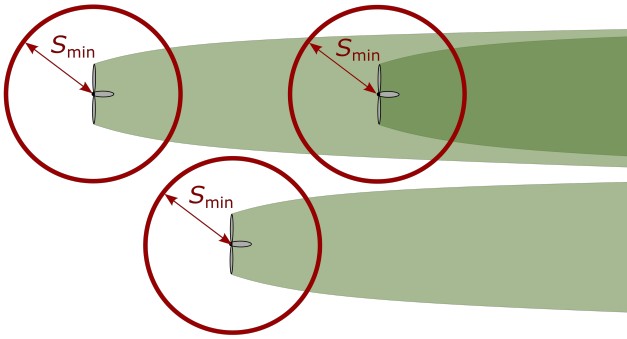

**Figure 13.** Schematic view of the safety space around three wind turbines to ensure a minimal turbine distance of $S_{\mathrm{min}}$. The shaded areas indicate the wake regions at a wind flow from left to right.

As for the validation, the wind farm Horns Rev 1 serves as reference case for the optimisation study. Therefore, the outer boundaries of HR1 are set as the optimiser's outer boundaries for turbine placement. The wind rose data is provided by Kirchner-Bossi and Porté-Agel (2018) and is shown in Fig. 14. The wind rose on the left-hand side is highly resolved and

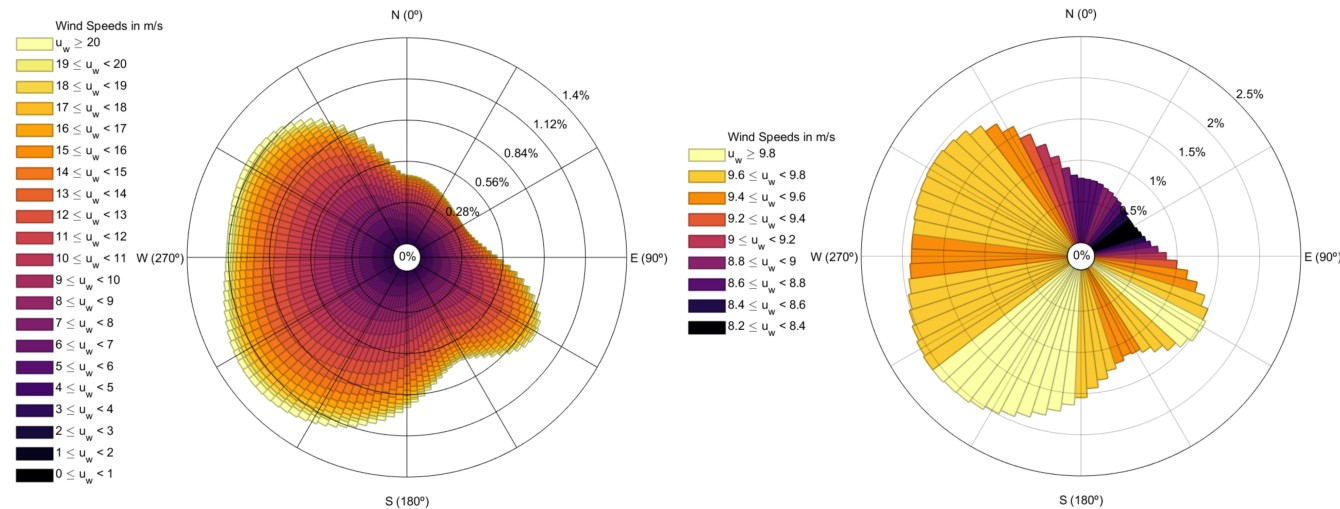

**Figure 14.** High resolved wind rose with 2640 combinations (left) and reduced one (right) with 72 bins, adapted from data by Kirchner-Bossi and Porté-Agel (2018).

consists of 2640 combinations of wind speeds and directions. As the optimisation problem is very time consuming, a reduced wind rose is proposed, as illustrated on the right-hand side of the graphic. Here, the wind directions are given in $5°$ bins, resulting in 72 directions with one single wind speed. Each sector of the reduced wind rose provides the same wind power and frequency as the corresponding bins of the resolved wind rose. The optimisation procedure itself is processed using the reduced wind rose. The optimised layout is then evaluated using the resolved wind data returning the optimised $AEP_{\mathrm{Farm}}$. Analogous





**Table 4.** Optimisation Key Data.

| Wind farm data | | |
|---|:---:|:---:|
| **Parameter** | **Symbol** | **Value** |
| Number of turbines | $N$ | 80 |
| Reference layout | $(X_i, Y_i)$ | Horns Rev 1 |
| Turbine model | - | Vestas V80 |
| Rotor diameter | $D_1$ | 80 m |
| Ambient turbulence | $TI_a$ | 7.7 % |
| **Optimisation settings** | | |
| *fmincon* algorithm | - | interior-point |
| Minimal turbine distance | $S_{\min}$ | 3.7 D |
| Outer boundaries | $(X_{\min,\max}, Y_{\min,\max})$ | Horns Rev 1 |
| **Variables** | $(X_i, Y_i)$ | farm layout |
| **Target quantity** | $AEP_{\mathrm{Farm}}$ | |

to the validation study, an ambient turbulence intensity of $TI_a = 7.7\%$ is chosen for all considered wind speeds following measurements by Hansen et al. (2012). To find an appropriate minimal turbine spacing the average length of the near wake region for the modelled conditions needs to be predicted. For that, the Ishihara-Qian wake model is run for all cases described in the wind roses. The rough, maximum occurring streamwise length for the near wake region is hereby estimated to be 3.7 D. Therefore, a minimal turbine distance of $S_{\min} = 3.7$ D is chosen as the final boundary condition. A summary of all settings 405    and conditions applied to the optimisation study can be seen in Table 4.

### 6.2 Optimisation results

This optimisation study contains two different optimisation cases. First, the layout is optimised using Qwyn with all models as described in Sect. 2. Then, to investigate the influence of the wake meandering model, a second optimisation is performed, where the wake meandering correction is turned off.

The result of the first case is shown in Fig. 15. The round marks indicate the turbine positions, while their colour visualises their individual $AEP_i$. The left-hand side graph shows the layout and energy yield of HR1 for reference. While the region in the centre of the farm shows a lower power yield, it can be seen that turbines at the edges are more productive. This is reasonable, since the centre region is most affected by wakes, while turbines at the farm's edges are more often operating under undisturbed conditions. Note, that the highest power of the farm is in its lower left corner, which corresponds to the shape of the wind rose



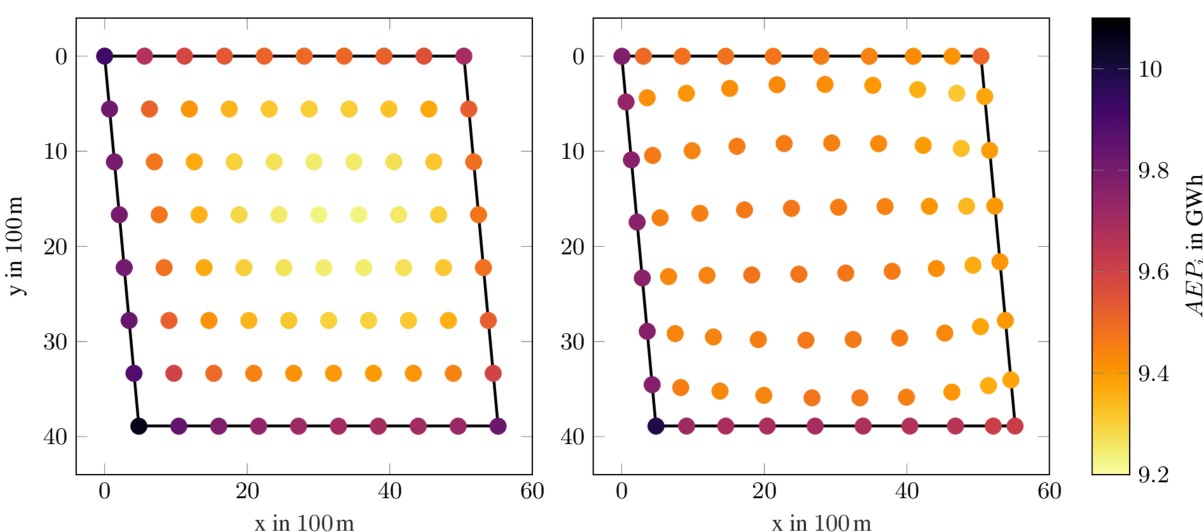

**Figure 15.** Optimisation result with reference (a) and optimised proposal (b). The coloured representation indicates the $AEP_i$ of the individual turbines.

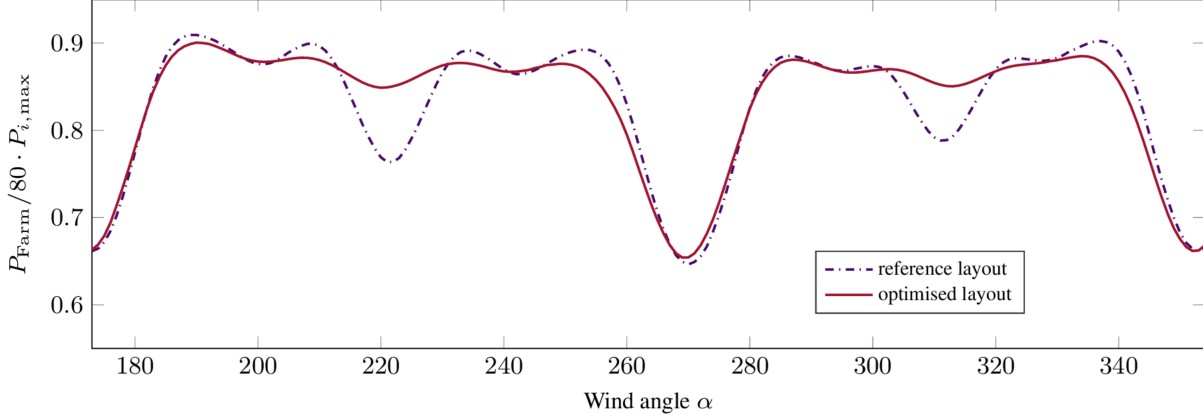

**Figure 16.** Normalised power output of Horns Rev 1 for a wide range of inflow angles. The graphic resembles the first optimisation case.

in Fig. 14. The right-hand side of Fig. 15 shows the optimised layout. The algorithm proposes a convex shape of the turbine rows, which increased the predicted power yield by $\Delta AEP_{\text{Farm}} = 0.12\,\%$. This results in the same order of magnitude found by Kirchner-Bossi and Porté-Agel (2018). The new layout shows a more even distribution of power yield in the centre of the farm. This is achieved at the expense of lower energy production at the farm's edges. Still, the left lower corner is showing the highest yield. A comparison of sensitivity to wind direction between the reference and optimised layout is shown in Fig. 16.

For continuity reasons, the graph shows a directional interval of $\Delta\alpha = [173°, 354°]$ as previously used in the validation study (cf. Fig. 11). The predicted power output of the farm is normalised by the output of 80 corresponding turbines operating in free stream conditions. The graph shows that the area under the curve increases after optimising the layout. Two significant power





drops are reduced at $\alpha_1 = 222°$ and $\alpha_2 = 312°$. At the same time, the curve is flattened for most other wind directions. This illustrates the optimisation's limitations in increasing the AEP, which is likely due to strict boundary conditions and high

turbine density. Four further power minima can be found at $\alpha = 90°$, $\alpha = 172°$, $\alpha = 270°$ and $\alpha = 352°$, where the first two are not shown in Fig. 16. Here, the predicted deficits in power are similar for both reference and optimised layout. This occurs due to the outer boundaries of the modelled domain. The turbines align alongside the wind farm's borders, since the

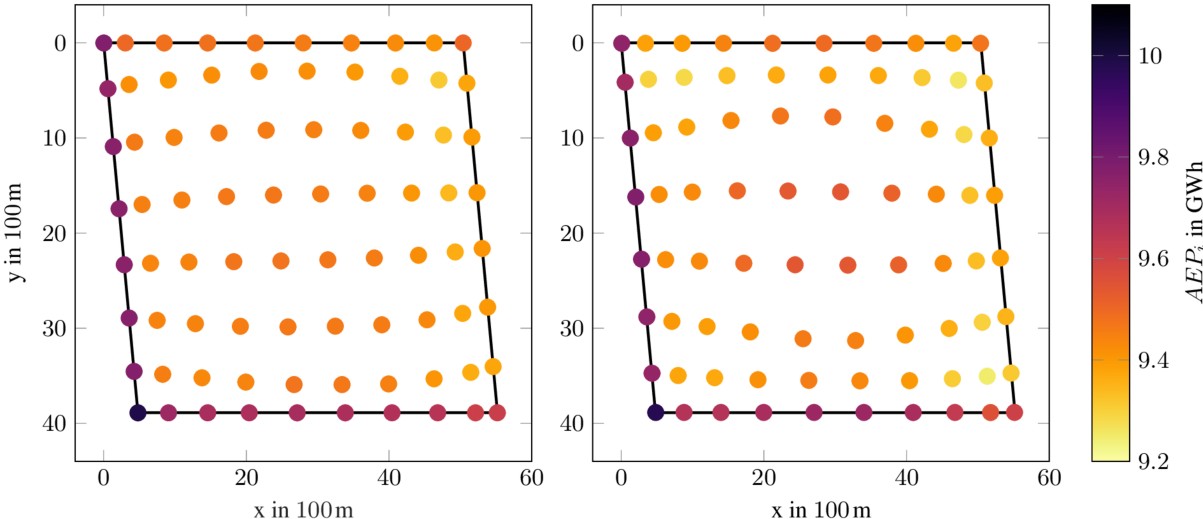

**Figure 17.** Comparison of optimisation results using wake meandering model (left) and without wake meandering model (right).

optimisation function can not violate these. As a result, four worst case wind directions remain, where the wakes align with downstream turbines, which can not be dampened due to the given constraints. This underlines the optimisation's restriction

by the given boundary conditions.

To evaluate the influence of wake meandering, a second optimisation is conducted without meandering correction. The achievable increase in annual energy production is $\Delta AEP_{\mathrm{Farm}} = 0.10\,\%$ and therefore very similar compared to the first result, as shown in Table 5. Figure 17 compares the first optimised layout with the second optimisation case. The optimisation

**Table 5.** Achieved increase in $AEP_{\mathrm{Farm}}$ with respect to the reference layout for both optimisations conducted.

| Modelling approach | $\Delta AEP_{\mathrm{Farm}}$ |
|---|---|
| Proposed tool | $+0.12\,\%$ |
| Without meandering correction | $+0.10\,\%$ |

function balances the power production in the centre of both proposed layouts by introducing a convex shape of the rows.

While layout adjustments for the meandering-neglecting version are more noticeable, the power yield improvement is similar when compared to the layout obtained including the meandering correction. Even though the power production for the second





optimisation case is balanced over the farm, we can observe lower yield not only at the very edges of the farm, but also at turbines located at the corners of the second rows. It appears that the optimisation has to adjust the layout more significant to achieve a comparable power gain. This can be traced back to the fact, that models without meandering correction overestimate

wake induced power losses as addressed in Sect. 5. More interestingly, when evaluating the second proposed layout with meandering correction, the wake losses exceed those of the reference case. A mindful choice of models and ambient parameters shows to be very important, to not worsen a designed layout, rather than optimising it.

As shown in Fig. 4, the inclusion of meandering correction in the model results in broader and less intense wakes. Because of that, larger parts of the wind farm's internal area are affected by wake effects. This fact is more realistic when thinking of the

real dynamic nature of the flow field, compared to conservative steady-state results. Consequently, parts of the farm's internal area that are little affected by wake effects become smaller. To relate this back to the optimisation: when areas of low wake effect become smaller, the power increasing relocation of turbines is more restricted. This leads to significantly lower possible layout adjustments, as observed.

While the study confirms, that considering wake meandering influences both prediction and optimisation of a wind farms

AEP, the results suggest that the reference layout (cf. Fig. 15) performs similar to the optimised proposals. It is important to mention that this study did not necessarily discover a global optimum of the problem. A handful of random starting points continuously returned the same result. However, choosing a Latin Hypercube sampling plan (Forrester et al., 2008) or deploying a sophisticated Kriging optimisation method (Kamaludeen et al., 2016) could return a global optimum with higher certainty. Furthermore, the results also show the limitations of the considered optimisation case. Varying constraints, such as the farm's

outer shape and type or number of turbines are likely to yield higher AEP improvements, than merely focusing on the distribution of turbines within a given area. The results show, that the optimiser tends to reduce linear arrangement of turbines by proposing a convex shape of rows. Investigating a circular shape of the farm could support this behaviour highlighting the true potential behind breaking up conservative layout design strategies. However, the feasibility of such a farm is of course bound to legal and contractual ties, which need to be further examined when suggesting an innovative farm shape. Another important

point to mention is the sensitivity of the optimisation to the given wind rose. As addressed in Sect. 6.1, the optimisation uses a wind rose, which is reduced to $5°$ bins with one wind speed each. The evaluation is then performed with the fully resolved data comprising 2640 combinations of wind directions and speed. When evaluating the optimised results with the reduced wind rose, the predicted power improvements from reference to optimised layout are roughly twice as high as the values summarised in Table 5. This emphasizes the importance of high quality ambient wind data when running AEP optimisation. Yet, the use

of reduced wind rose data is mainly motivated by the long computation time of the optimisation problem. Here, too, the introduction of a Kriging-surrogate optimisation method could further improve the tool's performance, decreasing the computation time (Kirchner-Bossi and Porté-Agel, 2018; Kamaludeen et al., 2016). This, in turn, could open up the opportunity to optimise farm layouts using higher resolved wind roses, thus, increasing the quality of the result.

Finally, the model shows a purely aerodynamic optimisation of the given problem. In reality, the partial overlapping of wakes

leads to an asymmetric load of the turbine rotors, increasing wear and maintenance. Also, the length of electrical cables is a





high-cost factor when planning wind farms. Therefore, a separate analysis of the turbine loads and costs must be done when designing and optimising wind farm layouts.

## 7  Conclusions

This work investigates an extended steady-state modelling approach for wind farms. The wakes' influence on the farm's power
output is predicted by using analytical modelling techniques joining them into Qwyn, a new computational framework. Qwyn incorporates the single-wake model by Ishihara and Qian (2018) as its foundation, the momentum-conserving superposition method by Zong and Porté-Agel (2020) for wake mixing, and a wake meandering correction based on work by Braunbehrens and Segalini (2019).

To verify the accuracy and reliability of Qwyn, extensive validation against measurement data, LES results and a conserva-
tive analytical modelling approach is conducted at the Horns Rev 1 wind farm. The results demonstrate the tool's capability to capture the behaviour and influence of wakes and their interactions on the wind farm's power output. The momentum-conserving superposition method improves the predicted results considerably with respect to the conservative approach of linear, rotor-based summation. When considering narrow wind bins of $\pm 1°$, Qwyn performs best when neglecting the wake meandering correction, yielding RSME values of down to $2.52\,\%$ with respect to the measurement data. However, for wider
bins of $\pm 5°$ the wake meandering correction adds significant value to the power prediction, where the minimum RSME captured is only $0.62\,\%$. Moreover, Qwyn performs reasonably well at different wind directions, capturing all trends in predicted farm power output by the LES validation data. This enables the prediction of a wind farm's annual energy production (AEP) when provided with the corresponding wind rose data.

The tool is then utilised to predict and optimise the AEP of HR1 using a whole wind rose. An optimised layout design
is suggested by embedding the tool into an optimisation structure using the gradient-based optimisation function *fmincon* in MATLAB. The study finds that an optimised, convexly shaped farm layout increases the predicted yearly power output by $0.12\,\%$, which corresponds to the magnitudes found by Kirchner-Bossi and Porté-Agel (2018). Morever, the study is extended to investigate the influence of wake meandering correction on the resulting optimised layout proposal. Omitting the wake meandering correction led to a very close increase in AEP of $0.10\,\%$. This however was achieved by a more distinct convex
layout shape, which underlines the importance of correct choice of models to obtain a realistical optimisation result. Even though Qwyn only considers a purely aerodynamic optimisation, the study shows potential for a layout improvement. At the same time, it exemplifies the optimisation's limitations to restrictive boundary conditions, which suggest further examination of farm area shape, as well as turbine model and density evaluation. This could be investigated in future studies, e.g. by chosing the energy density of a wind farm as optimisation target.

This work contributes to the field of fast and efficient wind farm modelling and optimisation by developing an analytical tool capable of representing aerodynamic interactions within wind farms. The insights gained through this research pave the road for further optimisation studies with focus on additional performance-defining parameters of a wind farm.



*Code and data availability.* Both MATLAB code as well as result data in this paper are available from the contact author.

*Author contributions.* DS: original study concept, programming & optimisation, literature review, data analysis, writing. JB: literature re-
view, conceptualisation, paper review; THH: supporting LES simulation & optimisation, paper review; GS: literature review, paper review.
All authors have contributed to and agree with the contents presented in this paper.

*Competing interests.* The contact author has declared that none of the authors has any competing interests.

*Disclaimer.* This work is based on the Masters thesis of Daniel Sukhman, from 2023 at the Technical University of Braunschweig. It has not
been published or under review in any other journal or similar publication.

*Acknowledgements.* The authors would like to extend their gratitude to Øyvind Waage Hanssen-Bauer (IFE) for sharing insightful literature,
to Nicolas Kirchner-Bossi (EPFL) for sharing the metrological wind roses, and to Jan Göing and Sebastian Lück from the Institute of Jet
Propulsion and Turbomachinery led by Prof. Jens Friedrichs at the Technical University Braunschweig for their invaluable support and
supervision through their extensive network and expertise.





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
