# Peer review of "Wind farm layout optimisation including meandering correction and momentum conserving superposition"

_Wind Energy Science, 2024_

## Referee Comment (RC2)

**Referee's comment to wes-2024-62**

**General comments**

The paper proposes a tool for wind farm EAP estimation and optimization which is based on previous works. Overall, the work is more an implementation (with some questionable changes) of existing models and methods and not novel enough to be considered a scientific contribution. There are several reasons why the paper should not be accepted:

1. There is therefore little novelty in the methodology presented.
2. The few adaptations done to the models, in particular to reduce the dimensionality from three to two and a proposed added TI model in presence of meandering present some flaws.
3. The results of the validations do not clearly proof any improvement compared to past models
4. The proposed optimized layout as such a small AEP gain compared to the unknown uncertainty of the model that would not be accepted by industry.

In particular, regarding point 2, the application of the flow equations at the hub-height plane is questionable. This is equivalent to assuming a 2D planar wake, which is however profoundly different from a real one which is 3D or, at least axisymmetric if shear is neglected. Ishihara's and Zong's model were not formulated and calibrated for a 2D wake, so their application here is flawed.

If instead the authors meant to model only the hub-height flow of a 3D (or axisymmetric) wake, then there may be an underlying fundamental theoretical mistake. For instance, Zong and Porte-Agel derive their model including the vertical dimension, thus integrating mass and momentum equations not only in $x$ and $y$, buy also in $z$. If one wants to evaluate the flow at hub height, they still need to integrate the equations in $x, y, z$ and then evaluate the resulting model at $z = 0$. Getting rid of one dimension before solving the flow equations is a gross error in partial differential equation theory. This can be shown as follows.

The continuity equation integrated over a volume spanning from freestream ($x = -\infty$) to the generic $x$, and in the spanwise region $y \in [-L, L]$ reads:

$$\int_{-L}^{L} \int_{\infty}^{x} \left( \frac{\partial u}{\partial x} + \frac{\partial v}{\partial y} + \frac{\partial w}{\partial z} \right) dx \, dy = \int_{-L}^{L} u(x,y)dy - 2LU_\infty + 2\int_{\infty}^{x} v(L,y)dx + \int_{-L}^{L} \int_{\infty}^{x} \frac{\partial w}{\partial z} dxdy = 0$$

The momentum equation in conservative form is:

$$\frac{\partial uu}{\partial x} + \frac{\partial uv}{\partial y} + \frac{\partial uw}{\partial z} = -\frac{\partial p}{\partial x} - |f| + \nabla \cdot \tau_x$$

Where $\tau_x$ are the turbulent and viscous stresses in x. Integrating over the volume yields:

$$\int_{-L}^{L} u^2(x,y)dy - 2LU_\infty^2 + 2U_\infty \int_{\infty}^{x} v(L,y)dx + \int_{-L}^{L} \int_{\infty}^{x} \frac{\partial uw}{\partial z} dxdy = -\int_{-L}^{L} |f|(y,z)dy$$

We here have used the Gauss' theorem and the fact that we assume $\tau_x$ at the boundaries of the domain to get rid of the stress term. Substituting the integrated continuity equation multiplied by $U_\infty$ finally gives:

$$\int_{-L}^{L} |f|(y,z)dy = U_\infty \int_{-L}^{L} u(x,y)dy - \int_{-L}^{L} u^2(x,y)dy + U_\infty \int_{-L}^{L} \int_{\infty}^{x} \frac{\partial w}{\partial z} dxdy - \int_{-L}^{L} \int_{\infty}^{x} \frac{\partial uw}{\partial z} dxdy$$

Or:

$$\int_{-L}^{L} |f|(y,z)dy = \int_{-L}^{L} \Delta u(x,y)u(x,y)dy + \int_{-L}^{L} \int_{\infty}^{x} \left( U_\infty \frac{\partial w}{\partial z} - \frac{\partial uw}{\partial z} \right) dxdy$$

Thus when considering just the hub height plane instead of $x, y$, and $z$ (like in Zong's model) or equivalently $x$ and $r$ (like Ishihara and Qian do) the equation changes significantly because:

- $\int_{-L}^{L} |f|(y,z)dy$ is not the thrust of whole the turbine but just its hub-height integral
- $U_\infty \int_{-L}^{L} \int_{\infty}^{x} \left( \frac{\partial w}{\partial z} - \frac{\partial uw}{\partial z} \right) dxdy$ appears and it is unknown a priori

An alternative approach to get rid of the z dimension is to perform a vertical average like in Letizia and Iungo, 2022. In that case, though, the formulation becomes still quite complex and includes dispersive stresses and vertical fluxes of mass and momentum that need to be modeled.

Moreover, the proposed meandering correction for added turbulence intensity based on the simple analogy with the velocity deficit is not justified sufficiently. The authors are missing what Keck et al. 2015 call "apparent turbulence intensity", which is due to the velocity fluctuations caused by meandering of the static (or stable, or MFOR) velocity deficit. This contribution is only a function of the wake shape (mostly its spanwise gradient) and adds up to the static turbulence intensity. The figure below shows a simple Montecarlo estimation of the meandering contribution, obtained by generating 1000 Ishihara-based velocity deficit and added TI profiles that meander base on a wake center which has a standard deviation in time of 0.25 D (left column). While the velocity deficit exhibits the expected spreading (right top), the total TI is even higher when meandering is introduced due to the apparent TI.

[Figure]

Regarding point 3, the claim that the proposed model does better than previous model is not supported by the evidence. In Fig 9 the proposed model does worse than the ones without meandering, while for larger averaging sectors the wake meandering seems to slightly improve results. This points to the fact that the meandering correction, which is smoothing the wake deficit, is likely not representing a physical effect, it is rather reducing the wake strength for larger wind direction bins and possibly leading to some error cancellation. If there was a physical effect being captured, results should improve regardless of the bins choice. It may also be that the way the experimental results are treated for narrow bind sector is somehow killing natural meandering, leading to the experimental results to be biased instead, but his has not being discussed. On the other hand, a model user does not know when and when not to use the meandering correction based on these results.

Other minor comments are provided below just for reference.

**Specific comments**

- L 10: The Frandsen model is not Gaussian but top-hat. You could add the Gauss-Hybrid-Curl instead [King et al, 2021].
- L 30: consider removing the statement on the Niayifar's models being the best since the Zong's model is used in the work and is again described as the most accurate at L 35.
- L 38: Medici et al. advocate for the internally-driven meandering hypothesis similar to the vortex shed by a cylinder, while Larsen et al., 2008 are the ones arguing that large atmospheric eddies cause meandering. Please amend this paragraph.
- L 46: "hubheight">>"hub height"
- L 50: if the Gaussian shape is assumed it cannot be derived from momentum equation. Momentum equation is instead used to calculate the depth and spread of the Gaussian based on the thrust. Please correct the sentence.
- Eq 1: $\phi$ is function of both x and y, since $\sigma(x)$, so either $\phi(x, y)$ or $\phi\left(\frac{y}{\sigma}\right)$. Please correct.
- Eq 4: the definition of $\phi$ should be placed right after Eq 1.
- Eq 13: same as Eq 1, $\phi$ is function of both x and y.
- Eq 15 and 16: the sign of the $y$ in the limit of $k_2$ should be negative because we are considering the lower side of the wake. The original formulation used $r$ which is always positive by in this case is different. The $k_1$ and $k_2$ should also be 0 if y is negative and positive, respectively, and also this condition is missing.
- L 120: the first and second sentences could be removed. There is no need to emphasize that all the models proposed before Zong are "unsound".
- Figures 5, 6 and 12 are not very informative, the algorithm follows the general "bin" method for AEP estimations and layout optimization presented in countless studies.
- Table 2: the experimental data also are collected based on bins, but from the table it seems that arbitrary 1 and 5-deg bins are used for the simulation instead but not for the experimental data analysis. Ther use of 8% and then 7.7% TI is unclear too.
- L 301: It is not clear which part of the meandering model is fitted to the data. The model shown in Section 2.3 seems to have all the constant predefined so it should be explicated which won has been changed to match the data.
- L 330: the explanation of the mismatch based on the wake rotation is not supported sufficiently. The reference by Qian and Ishihara. 2020, talks about the reduced turbulent production for partial wake overlapping but not based on rotation but on the magnitude of the velocity gradient. The wake rotation is indeed seen to decay quite fast behind the turbine even for low TI (Iungo et al, 2013, Fig. 2b).
- L 380: the expression "boundary conditions" is more suited for the solution of some PDE systems. In optimization context the term "constraint" is more common.
- L441: the optimization without meandering has higher wake losses than which "reference", the optimized with meandering or the real layout? And how can an optimization make the objective function worse?

**References**

Letizia, Stefano, and Giacomo Valerio Iungo. "Pseudo-2D RANS: A LiDAR-driven mid-fidelity model for simulations of wind farm flows." Journal of Renewable and Sustainable Energy 14.2 (2022).

King, Jennifer, et al. "Control-oriented model for secondary effects of wake steering." Wind Energy Science 6.3 (2021): 701-714.

Larsen, Gunner C., et al. "Wake meandering: a pragmatic approach." Wind Energy: An International Journal for Progress and Applications in Wind Power Conversion Technology 11.4 (2008): 377-395.

Keck, Rolf-Erik, et al. "Two improvements to the dynamic wake meandering model: including the effects of atmospheric shear on wake turbulence and incorporating turbulence build-up in a row of wind turbines." Wind Energy 18.1 (2015): 111-132.

Iungo, Giacomo Valerio, et al. "Linear stability analysis of wind turbine wakes performed on wind tunnel measurements." Journal of Fluid Mechanics 737 (2013): 499-526.

---

## Author Comment (AC2)

**Authors reply to 3nd referee's comments to wes-2024-62**

Thank you very much for your answer. We value your feedback and questions. We will answer any questions left open after reading our manuscript:

**General comment**

1. The main reason is the limitation of available computational resources. While the implementation of a "conservative" approach computes the AEP in fractions of seconds, adding the listed new methods slows the computation down significantly. This effect is negligible for one single AEP prediction. However, the duration of the optimization problem would be increased beyond tolerance. Ongoing work, which is the next step within this larger PhD project, includes the third dimension at reduced computation expense.

2. Those parameters are empirically fitted by the authors in the wake model's original publication [Ishihara & Qian, 2018]. The calibration was performed from a wind tunnel test, aerodynamic characteristics of a 2.4MW turbine, which is close to the Vest 2MW turbine used in this study, and a LES study. Further studies used by Ishihara & Qian to calibrate the parameters were LES simulations by Wu & Porté-Agel, 2011 & 2012.

3. In these LES studies Porté-Agel, 2013 and Wu & Porté-Agel, 2015, performed simulations of the wind farm Horns Rev 1 under neutral atmospheric conditions with $u_\infty = 8\frac{m}{s}, TI = 7.7\%$ and varying wind directions. The information is given in paragraph 4 "Study case". These boundary conditions apply for both studies, not just the study performed in 2015. We do agree that this should be made more clear in the manuscript and will add a sentence accordingly.

4. The reference used here is the wind farm modelled with Qwyn in its original layout, both with and without wake meandering. Figure 16 only depicts the case modelled with wake meandering. The black dashed line shows the modelled power output of the reference layout over various inflow angles. The optimization study does not include comparisons to measurement or LES data. We will clarify this with an additional sentence in the introduction paragraph of the optimization chapter.

   The estimated AEP is 758.62 GWh with meandering and 758.52 GWh without meandering. We emphasize that the absolute AEP value is less informative. While we use a relatively high-resolution of 2640 combinations of wind direction and speed, Kirchner-Bossi & Porté-Agel (2018) mention that the correct overall AEP prediction depends on the resolution of the atmospheric data. The result will therefore be refined, when the input of atmospheric data is refined.

   We agree, that the tool overpredicts the power output for most wind directions. This overestimation is likely due to the reduction of our modelling approach to two dimensions, which introduces errors in modelling partially overlapping wake cases. The wake shape is modelled more in a rectangular or linear rather than its actual circular shape. We will elaborate on this in the methodology section of the momentum-conserving superposition and the conclusion of the validation chapters.

**Specific comments**

1. The calibration and derivation of the equations has been performed by Ishihara & Qian in their original publication. We would like to refer to our answer of General comment No.2. We agree, that simply referring to the original publication of the model is not enough. In the second draft we will elaborate the model's calibration and refer to the original publication & it's references in a few additional sentences.

2. Ishihara & Qian do not specifically define a near and far wake region. They rather develop a model which is applicable in both far and near wake. The empirical parameters *a* and *b* are fitted to the far wake region with the use of wind tunnel & full-scale measurement data as well as LES simulations as discussed in "General comment: Answer 2". In the Ishihara-Qian (2018) wake model, there is no explicit mathematical definition for the near and far wake regions. The model incorporates several empirical parameters fitted to measurement and LES data. The parameters $a$ and $b$ (Eq. 6) are fitted to the far wake region, while parameter $c$ (Eq. 8) is fitted to the near wake region, with its influence decaying due to the $(1 + x/D)^{-2}$ term in Eq. 7. Ishihara and Qian do not provide specific details on how they quantitatively distinguish between near and far wake regions. However, they describe the far wake region as where "the wake is fully developed and the velocity deficit and the added turbulence intensity can be assumed axisymmetric and have self-similar distributions in the wake cross-sections" (Ishihara & Qian, 2018), as noted by Vermeer et al. (2003). We, therefore, assume that the original authors evaluated measurement and LES data to determine the regions for fitting parameters $a$, $b$ and $c$.

   For completeness: Qwyn incorporates a model extension for yawed wind turbines (wake deflection model) introduced by the same authors in Qian & Ishihara (2018). Here, Qian & Ishihara propose a quantitative estimation of near and far wake regions based on turbine yaw. This does not apply to any cases considered in this study, which is why we adhere to the previous explanation.

   It is also important to note that the near wake region does not play a significant role in the scope of the present study.

   We hope this clarifies the definition of near and far wake regions in the given approach.

3. This question aligns with the one under "general comments Nr.1". As mentioned, computational resources are the main driver for computing information only in a two-dimensional plane. Consequently, the integration can only be done in two dimensions. We are aware, that this introduces an error, which we will address more clearly in the second draft of the manuscript. We will provide additional explanations within the methodology section of the Zong – Porté-Agel model and in the validation conclusions. Despite this simplification, our reduced dimension model shows good agreement with the validation data, giving us confidence in the study's results. However, we plan to avoid this dimensional simplification in future work.

4. We define the worst-case operation scenarios as those with the most severe wake losses, occurring when a high number of turbines are aligned with the wind direction (particularly at 270° and symmetrically at 90°). In these scenarios, overlapping wakes significantly reduce the rate of wake recovery within the farm, leading to notable drops in performance, as seen in Figure 11. The most severe drop is at 270°, detailed in Figure 9, where there is

an overall loss of approximately 40% for all turbines behind the first one. To provide context, we propose citing Barthelmie et al., 2009, who demonstrated that these drops are most distinct when a large number of wakes align.

5. This is correct. We will clarify this in our manuscript.

6. The uncertainties are either quantitatively given in the cited reference (wind speed $\pm 0.5$ m/s) or estimated by us through evaluating the error bars provided in the source's graphs (Turbulence intensity $\pm 2\%$ and normalized power $\pm 0.1$).

7. We believe that the observed differences are not due to the impact of wake meandering but rather indicate the limitations of the meandering correction model. Braunbehrens & Segalini (2019) validated their model on wide directional sectors, showing good results, which we were able to reproduce in our validation. However, when applied to narrow bins, the model performs significantly worse. This variability can make it challenging for users to determine when the meandering correction adds value. Therefore, we will conclude more decisively in the second draft that the correction is not universally applicable.

8. This is correct. We will revise this sentence for clarity. We have not plotted the graph with 5-degree bins to align with the available LES data, which does not include data points at the resolution necessary for such binning. Consequently, we cannot compare it to a high-fidelity study or measurement data at this time. However, we agree that conducting such an analysis would be valuable for future research.

**Technical comments**

Thank you for pointing out those slips (especially point 3). We will correct these in the second draft.

**References:**

Ishihara, T., Qian, G.-W., 2018 "A new Gaussian-based analytical wake model for wind turbines considering ambient turbulence intensities and thrust coefficient effects." Journal of Wind Engineering & Industrial Aerodynamics. 177, 275-292.

Wu, Y.T., Porté-Agel, F., 2011. "Large-eddy simulation of wind-turbine wakes: evaluation of turbine parametrisations." Boundary-Layer Meteorology. 138, 345–366.

Wu, Y.T., Porté-Agel, F., 2012. "Atmospheric turbulence effects on wind-turbine wakes: an LES study." Energies 5, 5340–5362. Xie, S.B., Archer, C., 2014. Self-similarity and turbulence characteristics of wind turbine wakes via large-eddy simulation. Wind Energy 17, 657–669.

Kirchner-Bossi, N. and Porté-Agel, F., 2018: "Realistic Wind Farm Layout Optimization through Genetic Algorithms Using a Gaussian Wake Model". Energies, 11, 3268.

Vermeer, L.J., Sørensen, J.N., Crespo, A., 2003. "Wind turbine wake aerodynamics. " Progress in Aerospace Sciences. 39, 467–510.

Qian, G.-W., Ishihara, T., 2018 "A New Analytical Wake Model for Yawed Wind Turbines." Energies. 11(3), 665-689.

Barthelmie, R. J., Hansen, K., Frandsen, S. T., Rathmann, O., Schepers, J. G., Schlez, W., Phillips, J., Rados, K., Zervos, A., Politis, E. S., and Chaviaropoulos, P. K., 2009: "Modelling and measuring flow and wind turbine wakes in large wind farms offshore." Wind Energy, 12, 431–444.

Braunbehrens, R. and Segalini, A., 2019: "A statistical model for wake meandering behind wind turbines". Journal of Wind Engineering and Industrial Aerodynamics, 193, 103954.

---

## Author Comment (AC3)

**Authors reply to 2nd referee's comments to wes-2024-62**

Thank you for your thorough and thoughtful review. The authors are grateful for your valuable insights and constructive criticism. We believe that the provided clarifications and referenced literature will significantly help us in refining our manuscript and addressing points that were not well explained. Your feedback is instrumental in improving the quality and clarity of our work, and we appreciate the opportunity to enhance our study accordingly.

The referee introduced four major points as to why the paper should not be accepted.
1. Flawed model adaptation
2. Validation does not proof any improvement
3. Novelty
4. Optimized layout would not be accepted by industry

In the following sections, the authors will reply to these points. Note that the order of the points named is the order of replies and not the original order as mentioned by the referee. Lastly the authors will answer to the specific comments in Section 5.

**1. Flawed model adaptation**

The presented code solves analytical explicit equations, utilizing the advantage of lower computational resources when compared to any CFD application or code that solves PDE's. The authors indeed apply the flow equation at a hub height plane, as assumed in the first paragraph of the referee's explanation, rather than modelling a 3D flow. The main reasoning behind this is the reduction of computational time, making the code applicable for a preliminary optimization study. Eliminating the z-dimension when solving PDE's is indeed a significant error, as the referee points out. However, our approach focuses on solving simplified, explicit expressions and not PDE's. This type of modelling is specifically chosen to reduce computational time.

We would like to address the application of Ishihara's and Zong's models separately, as there is a fundamental difference in the influence of the third dimension on the model's results. We will then address the integration error introduced by the neglect of the height dimension when combining both models.

*Ishihara - Qian*
Regarding Ishihara's model, the equations are solved for specific points behind the turbines to compute flow velocity and rotor-added turbulence intensity. Based on the PDE's derived from momentum conservation, these equations form an explicit expression describing the wake region as a self-similar, Gaussian-shaped area. This allows for the prediction of flow properties at any point behind the turbine (in streamwise $x$, spanwise $y$, and height $z$ direction) independently of other points. While this simplified approach neglects external pressure gradients and forces to maintain the Gaussian assumption [Appendix A., Ishihara-Qian, 2018], it significantly reduces computational expense. Consequently, the model's application is limited for conditions with low vertical mixing and shear, such as neutral to stable atmospheric conditions and for environments with low surface roughness, which is typical in offshore settings. Even though the model can predict all three spatial dimensions, it is up to the user to choose which points in the wake region to compute. For example, if one would like to depict the wake region in the $x - z$ plane, one would compute all necessary points $(x_i, z_i)$. When information at hub height is needed, all necessary points in the $x - y$ plane at $z = 0$ can be computed without considering other points in the vertical direction, as they are not interdependent in explicit models.

For such a case, Ishihara's equations are solved for

$$r = \sqrt{y^2 + (z - H)^2} \; ,$$

where $z = H$ leading to

$$r = |y| \, .$$

This point was not sufficiently clarified in the submitted manuscript, which leads to confusion for the definition of $y$ in the submission's equations. This will be pointed out clearly in the paper's second draft in chapter "2.1 Ishihara-Qian wake model".

Neglecting the third dimension during the computation of the turbine's individual wakes does not lead to any error or additional calibration uncertainties, as it is merely a decision on which points to compute, distinct from the integration of inflow parameters, which will be addressed separately.

*Zong - Porté-Agel*
Regarding the Zong superposition model, the question of simplification is more complex. The model's derivation includes several assumptions that initially reduce the problem's complexity. Apart from assuming uniform inflow and neglecting the Reynolds normal stress term $\langle u'u' \rangle$, the model by default assumes no turbulent momentum transfer in vertical and spanwise direction ($\langle u'v' \rangle \approx 0, \langle u'w' \rangle \approx 0$) [Appendix A, Zong, Porté-Agel, 2020]. This leaves us with the following equation for the integrated streamwise momentum

$$\rho \iint_{Inlet} u_\infty \, \mathrm{d}y\mathrm{d}z - \rho \iint_{Outlet} u_i(x, y, z)^2 \, \mathrm{d}y\mathrm{d}z = F_T \; ,$$

where $u_i$ is the velocity in the wake and $F_T$ is the thrust force induced on the flow. Combining with the conservation of mass, the thrust force equation changes to

$$F_T = \rho \iint_{Outlet} u_i(x, y, z) \cdot \Delta u(x, y, z) \, \mathrm{d}y\mathrm{d}z \, .$$

Here, $\Delta u(x, y, z)$ is the velocity deficit in the wake, defined as $\Delta u = u_\infty - u_i$ [$Zong, Porté - Agel, 2020$]. This derivation considers all three spatial dimensions while focusing on conserving the streamwise momentum, which is fundamental for this work.

Following Zong's derivations further, the momentum conserved properties of a mixed wake region is found by performing a weighted summation of the individual wakes

$$\Delta U_{Farm}(x, y, z) = \sum \frac{u_{c,i}(x)}{U_{c,Farm}(x)} \Delta u_{.}(x, y, z) \, .$$

Here, each individual wake $u_i(x, y, z)$ is linearized to a convection velocity $u_{c,i}(x)$. This individual wake's convection velocity can be derived from explicit equations such as the Ishihara or Bastankhah model, again leading to a simplified yet quick prediction. The question of dimensionality and neglection of the vertical $z$-direction becomes relevant when investigating the last entity of this model, the convection velocity of the farm's wake. Its original definition is given by

$$U_{c,Farm}(x) = \frac{\iint U_i(x, y, z) \cdot \Delta U(x, y, z) \, \mathrm{d}y\mathrm{d}z}{\iint \Delta U(x, y, z) \, \mathrm{d}y\mathrm{d}z} \; ,$$

where $U_i$ and $\Delta U$ are the reduced velocity and velocity deficit in the mixed wake within or behind the wind farm. When applying this equation to a discretized space, we express the integrals with the Riemann sum as

$$U_{c,Farm}(x) = \frac{\sum \sum U_i(x,y,z) \cdot \Delta U(x,y,z) \, \Delta y \Delta z}{\sum \sum \Delta U(x,y,z) \, \Delta y \Delta z} \, ,$$

which can be reduced to

$$U_{c,Farm}(x) = \frac{\sum U_i(x,y,z) \cdot \Delta U(x,y,z)}{\sum \Delta U(x,y,z)} \, .$$

This work reduces the computed information further to two dimensions, leading to

$$U_{c,Farm}(x) = \frac{\sum U_i(x,y) \cdot \Delta U(x,y)}{\sum \Delta U(x,y)} \, .$$

We, therefore, neglect velocity gradients in the vertical direction. In cases of unstable atmospheric stability, high surface roughness, or different turbine heights within a farm or cluster, vertical flux is significant. For those scenarios, the proposed simplification would lead to substantial errors as it cannot capture the wind's shear profile and vertical flux. However, the study presented in this manuscript does not involve such complex cases; we model one wind farm in offshore, neutral atmospheric conditions.

Regarding the integration of the turbines' performance parameters: to obtain the thrust and power of the rotor, the modelled velocity in the rotor plane is averaged to approximate the scalar inflow velocity. This value is then used to determine power and thrust through performance curves, as provided in the manuscript. We acknowledge that there is a substantial difference between evaluating the inflow wind speed in a two-dimensional plane versus a three-dimensional space. Graphically, as shown in Figure 1, the flow for the 2D case is approximated by a line of points at hub height. Averaging these points results in a rectangular representation of the flow, while the three-dimensional version would resolve the circular shape of the wake.

[Figure]

Figure 1: Points where flow properties are computed. Left with resolving the *z* dimension. Right, for resolving only a 2D plane at hub height.

This simplification can introduce errors, especially in cases with high partial wake overlap. The authors acknowledge this limitation and do not claim that the proposed 2D version of Qwyn is superior to other implementations that include three-dimensional space. However, focusing on Qwyn's combination of Ishihara's and Zong's models, our tool produces results that are comparable to conservative approaches while significantly reducing computational load (the effect of the wake meandering correction will be discussed separately). This is particularly noteworthy given the simplifications made to the model. Nevertheless, as this is ongoing work within the scope of a larger PhD project, the simplifications presented here are temporary. The next version of Qwyn will include the third dimension to eliminate potential errors and enable the modelling of more complex atmospheric stability and topology cases.

However, we agree, that this fact should be addressed and discussed in the paper. We will add an additional paragraph at the end of chapter "2.2 Momentum-conserving wake superposition", discussing this very limitation. It will also become important for the discussion of the results and will be mentioned in our answer to the referee's specific comment on "L. 330").

*Meandering correction (applied to the rotor-added turbulence)*
We do not claim to propose a fully accurate turbulence-meandering model. Instead, we are very grateful for the referee's extended insight into the field of rotor-added turbulence under meandering conditions. The development and implementation of an explicit formulation of rotor-added turbulence under these conditions could be the focus of future work, contributing to the field of rapid wake modelling.

In the scope of this work, we applied the used analogy because not modelling the rotor-added turbulence worsened the results consistently, highlighting the importance of turbulence modelling. This can be seen as a first step, and further improvements and implementations are made possible by the valuable information and citations provided in this review. We will mention this in the paragraph on future work in the manuscript's conclusions.

**2. Validation does not proof any improvement**

The green curve in Figures 9 and 10 of the manuscript (Momentum conserving superposition) is also obtained using Qwyn with the discussed 2D simplification. Despite reducing from 3 to 2 dimensions, the implemented framework can follow the trends of all other models and reduce the RMSE with respect to the linear rotor-based summation.

We tested the influence of different models on wind farm modelling, and one significant finding was that introducing the wake meandering model to the framework causes substantial deviations for narrow bins. The meandering correction is validated on cases based on wide bins [Braunbehrens & Segalini, 2019]. We agree with the referee that it is challenging to understand for which cases the meandering model improves results and for which it does not. In the one application case modelled (Horns Rev 1), the proposed model seems to improve results only when considering large averaging sectors, which could be due to numerical issues or error cancellation, as mentioned by the reviewer. However, such a claim cannot be made without further investigation. At present, we observe that the application of the meandering model is not universally reliable. This finding will be explicitly clarified in the conclusions of the validation chapter. Additionally, the fact that Qwyn, as a framework, is not limited to the use of the meandering correction will be clearly stated in the introduction of the chapter. We greatly appreciate the feedback and will incorporate these comments.

**3. Question of novelty**

This research does not claim to present a new model, but rather presents a novel computational framework. It effectively captures the power production trends of an offshore wind farm under various inflow conditions (wind direction, turbulence intensity, and speed). The presented computational framework makes it possible to test different models and their influence on layout optimization. To the best of our knowledge, no existing analytical, steady-state framework has included and tested a full wind farm with momentum-conserving superposition approach with wake meandering correction.

By implementing and testing these models (and simplifying them as shown), we have identified some limitations in the meandering correction, and demonstrated that reducing the model to a two-dimensional plane still yields reasonable results. This, along with the optimization study, provides valuable contributions to the field of rapid, simplified wake modelling that differ from the approaches in other studies.

Moreover, the observed influence of the wake meandering correction on a layout optimisation can be regarded as a novel contribution. It is not intuitively clear that the convex shape increases when neglecting meandering correction, highlighting the insights provided by our study.

**4. Optimized layout would not be accepted by industry**

The referee mentions that the optimized result presented in this work is too small to be accepted by industry. We believe this point may have been misunderstood. The optimization study never aimed to propose a new layout for the industry, especially not using an existing, outdated wind farm as its basis. Instead, the objective was to explore the potential of such optimization within the constraints of an existing wind farm.

The results indicate that the increase in power is minimal across all different modelling approaches, suggesting a need to broaden the scope of optimization parameters, as mentioned in the manuscript. Optimizing a layout within a predefined field and a fixed number of turbines yields only a minimal improvement in AEP. Our findings align with the optimization results in the cited literature, reinforcing the applicability of the given framework and supporting the idea that further research and expansion of parameters, such as farm shape, number, and size of turbines, are necessary. This insight alone is valuable for the industry, indicating that there may be more effective methods to optimize production than layout optimization within an already defined farm shape. We agree that this point has not been clarified sufficiently. We appreciate the referee's feedback, which will help us to refine and clarify the goals and implications of our study. The second draft of the manuscript will address this matter in the discussion & conclusion chapters.

**5. Answers to the specific comments**

- L 10: The Frandsen model is not Gaussian but top hat. You could add the Gauss-Hybrid-Curl instead [King et al, 2021].
  A: Thank you for pointing out that oversight. We will adjust this in the manuscript.

- L 30: consider removing the statement on the Niayifar's models being the best since the Zong's model is used in the work and is again described as the most accurate at L 35.
  A: We agree and will reformulate the statement to clearly state that Niayifar's superposition is the best among those not derived from mass and momentum conservation, which explains why it is used as the "conservative approach" in the scope of this study.

- L 38: Medici et al. advocate for the internally driven meandering hypothesis similar to the vortex shed by a cylinder, while Larsen et al., 2008 are the ones arguing that large atmospheric eddies cause meandering. Please amend this paragraph.
  A: We apologize for the oversight. While Medici et al. did observe wake meandering in their wind tunnel testing campaign, it was indeed Larsen et al. who not only proposed a DWM but also linked the meandering to "large scale air movements". We will correct this in our manuscript.

- L 46: "hubheight" >> "hub height"
  A: Will be corrected.

- L 50: if the Gaussian shape is assumed it cannot be derived from momentum equation. Momentum equation is instead used to calculate the depth and spread of the Gaussian based on the thrust. Please correct the sentence.
  A: We agree that our formulation can be misleading. The Gaussian shape itself cannot be derived from momentum conservation, as it is an assumption giving the explicit simplification of the PDE derived from the momentum equation. However, the assumption of the explicit equation (in Ishihara-Qian's case) is *based* on a PDE derived from the momentum equation. This distinction is crucial, as the application of Zong's model is limited to explicit models that meet this requirement. We will clarify this in the revised sentence.

- Eq 1: $\phi$ is function of both x and y, since $\sigma(x)$, so either $\phi(x,y)$ or $\phi(\frac{y}{\sigma})$. Please correct.
  A: Will be corrected.

- Eq 4: the definition of $\phi$ should be placed right after Eq 1.
  A: We will move Eq.4 and adjust the text.

- Eq 13: same as Eq 1, $\phi$ is function of both x and y.
  A: Will be corrected.

- Eq 15 and 16: the sign of the $y$ in the limit of $k_2$ should be negative because we are considering the lower side of the wake. The original formulation used $r$ which is always positive by in this case is different. The $k_1$ and $k_2$ should also be 0 if y is negative and positive, respectively, and also this condition is missing.
  A: Here, we would like to refer to our answer regarding the major comment: "Flawed adaptation – *Ishihara - Qian*". In our notation $y$ is used as $r$, where $r = |y|$. This will be clarified in the manuscript, to prevent confusion.

- L 120: the first and second sentences could be removed. There is no need to emphasize that all the models proposed before Zong are "unsound".
  A: It is not the authors intention to devalue any model or work before Zong. We will choose a softer formulation in this part of the text, as we believe it serves a good introduction to the motivation of using Zong's model.

- Figures 5, 6 and 12 are not very informative, the algorithm follows the general "bin" method for AEP estimations and layout optimization presented in countless studies.
  A: We agree that the information provided in the named figures is showcasing the basic structure of the computation framework. However, we would like to mention three reasons to show this computation structure in the paper:
  1) Depending on the readers background, the threshold for delving into this topic can be high, making such publications less inclusive.
  2) The general algorithm structure is necessary to ensure comparability & reproducibility.
  3) As a follow-up to the second point: ongoing improvements to the framework, which will be shown in future studies, can be directly compared and linked to this initial publication.

- Table 2: the experimental data also are collected based on bins, but from the table it seems that arbitrary 1 and 5-deg bins are used for the simulation instead but not for the experimental data analysis. Ther use of 8% and then 7.7% TI is unclear too.
  A: The measurement data is also binned in 1 and 5 deg. bins. This is a misunderstanding/ not sufficiently clarified in the table. We will adjust this. Regarding the turbulence: The LES data use 7.7% for all simulations. We decided to keep to it with our analytical modelling. The 8% are taken from the measurement data and are provided purely as additional information. We propose to remove this value to avoid possible confusion.

- L 301: It is not clear which part of the meandering model is fitted to the data. The model shown in Section 2.3 seems to have all the constant predefined so it should be explicated which won has been changed to match the data.
  A: Here, the manuscript is inaccurate. The authors of the manuscript have not altered/ fitted any of the constants to measurement data themselves. As pointed out correctly, all constants are predefined. What needs to be communicated here, is, that the meandering model has been *validated* with a 5-degree binned dataset (on Horns Rev) in the original publication by Braunbehrens & Segalini, 2020. We will correct this in the manuscript.

- L 330: the explanation of the mismatch based on the wake rotation is not supported sufficiently. The reference by Qian and Ishihara. 2020, talks about the reduced turbulent production for partial wake overlapping but not based on rotation but on the magnitude of the velocity gradient. The wake rotation is indeed seen to decay quite fast behind the turbine even for low TI (Iungo et al, 2013, Fig. 2b).
  A: That is a very important clarification & correction, thank you for providing insightful literature. This needs to be corrected in the given manuscript and discussion. It seems that it is rather the simplification to 2 dimensions, which drives the deviation between the presented modelling approach and the LES study, as discussed and displayed in the answer to the major comments: "Flawed adaptation: *Zong – Porté-Agel*" - Figure *1*.

- L 380: the expression "boundary conditions" is more suited for the solution of some PDE systems. In optimization context the term "constraint" is more common.
  A: Will be corrected.

- L441: the optimization without meandering has higher wake losses than which "reference", the optimized with meandering or the real layout? And how can an optimization make the objective function worse.

  A: The second proposed layout is found by running the optimisation while using Qwyn with the meandering model turned off. It does not make the objective function worse, but rather shows an insignificant change of AEP (minimally improved by 0.10%), as summarized in Table 5 of the manuscript. What is meant by the statement in L441 is, that when we evaluate this very layout with the meandering model turned on (thus – different modelling approach), we observe, that the AEP is not improved, but worsened with respect to the reference case, which in this case is also evaluated with the meandering correction turned on. We see therefore, that the choice of models influences the optimization results. At the same time all different modelling approaches yield an insignificant improvement in AEP, showcasing the limitation of an optimisation based on pure turbine placement adjustments in an already predefined outer farm shape.